# Epithelial expression of *Gata4* and *Sox2* regulates specification of the squamous–columnar junction via MAPK/ERK signaling in mice

Nao Sankoda[1,2,3], Wataru Tanabe[1,4], Akito Tanaka[1], Hirofumi Shibata[5], Knut Woltjen[1,6], Tsutomu Chiba[4], Hironori Haga[7], Yoshiharu Sakai[2], Masaki Mandai[8], Takuya Yamamoto [1,9,10,11], Yasuhiro Yamada [3,9], Shinji Uemoto[2] & Yoshiya Kawaguchi [1✉]

The squamous–columnar junction (SCJ) is a boundary consisting of precisely positioned transitional epithelium between the squamous and columnar epithelium. Transitional epithelium is a hotspot for precancerous lesions, and is therefore clinically important; however, the origins and physiological properties of transitional epithelium have not been fully elucidated. Here, by using mouse genetics, lineage tracing, and organoid culture, we examine the development of the SCJ in the mouse stomach, and thus define the unique features of transitional epithelium. We find that two transcription factors, encoded by *Sox2* and *Gata4*, specify primitive transitional epithelium into squamous and columnar epithelium. The proximal–distal segregation of *Sox2* and *Gata4* expression establishes the boundary of the unspecified transitional epithelium between committed squamous and columnar epithelium. Mechanistically, *Gata4*-mediated expression of the morphogen *Fgf10* in the distal stomach and *Sox2*-mediated *Fgfr2* expression in the proximal stomach induce the intermediate regional activation of MAPK/ERK, which prevents the differentiation of transitional epithelial cells within the SCJ boundary. Our results have implications for tissue regeneration and tumorigenesis, which are related to the SCJ.

[1] Department of Life Science Frontiers, Center for iPS Cell Research and Application (CiRA), Kyoto University, Kyoto 606-8507, Japan. [2] Department of Surgery, Kyoto University Graduate School of Medicine, Kyoto 606-8507, Japan. [3] Division of Stem Cell Pathology, Center for Experimental Medicine and Systems Biology, Institute of Medical Science, University of Tokyo, Tokyo 108-8639, Japan. [4] Department of Gastroenterology, Kyoto University Graduate School of Medicine, Kyoto 606-8507, Japan. [5] Department of Otolaryngology, Gifu University Graduate School of Medicine, Gifu 501-1194, Japan. [6] Hakubi Center for Advanced Research, Kyoto University, Kyoto 606-8501, Japan. [7] Department of Diagnostic Pathology, Kyoto University Hospital, Kyoto 606-8507, Japan. [8] Department of Gynecology and Obstetrics, Kyoto University Hospital, Kyoto 606-8507, Japan. [9] AMED-CREST, AMED 1-7-1 Otemachi, Chiyodaku, Tokyo 100-0004, Japan. [10] Institute for the Advanced Study of Human Biology (WPI-ASHBi), Kyoto University, Yoshida-Konoe-cho, Sakyo-ku, Kyoto 606-8501, Japan. [11] Medical-risk Avoidance Based on iPS Cells Team, RIKEN Center for Advanced Intelligence Project (AIP), Kyoto 606-8507, Japan. ✉email: yoshiyak@cira.kyoto-u.ac.jp

Boundaries separating neighboring tissues with distinct functions are required for proper organogenesis and tissue homeostasis[1]. The squamous–columnar junction (SCJ) in mammals, e.g., the esophageal–gastric junction and uterine cervix, is one of the boundaries that partition functionally distinct epithelial types[2]. The squamous epithelium serves as a strong barrier against mechanical stress, whereas columnar epithelium produces mucins to protect epithelial cells from external stimuli such as acid (e.g., gastric and bile acid) and invading microorganisms. Because the SCJ is located between functionally distinct epithelial types, the cells adjacent to it are exposed to stresses from the two distinct environments. Indeed, esophageal ulcers caused by mechanical stress or gastroesophageal reflux often occur at the junction between the esophagus and stomach, and these lesions need to be repaired[3]. Additionally, SCJs are hot spots for metaplastic lesions, e.g., intestinal metaplasia in the esophageal–gastric junction and squamous metaplasia in the uterine cervix[4,5]. Metaplastic lesions are adaptive states that form in response to abnormal stimuli, and often give rise to carcinomas when the stimuli persist for long periods of time[6,7]. Accordingly, stem-like cells have been implicated in the regeneration and tumorigenesis of damaged SCJs[8–10].

Recent studies showed that a KRT7-expressing transitional epithelium exists between the squamous and columnar epithelium at the esophageal–gastric junction and uterine cervix[11,12]. Furthermore, genetic manipulation of the cells constituting the transitional epithelium in mouse models and human organoid suggests that these cells could be the origin of metaplasia at SCJs[13]. Taken together, these observations raise the possibility that transitional epithelial cells have stem-like properties to maintain homeostasis of SCJs[10,13].

Although the involvement of the transitional epithelial cells in regeneration and tumorigenesis related to SCJs is proposed, the origins and physiological properties of transitional epithelial cells remain to be clear. During development, the SCJ is established at the later embryonic stage, when squamous and columnar epithelium are differentiated from a common pseudostratified epithelium[14,15]. A previous study suggested that the embryonic pseudostratified epithelium resides in adult transitional epithelium, and also proposed the possibility that the residual embryonic cells are the origin of Barrett's metaplasia[12]. In addition, key developmental genes regulating the regionalization of a gastro-intestinal tract are aberrantly expressed in the Barrett's metaplasia[16–18]. These observations prompted us to examine the development of transitional epithelium. In this study, using mouse genetics, lineage tracing, and organoid culture, we define the unspecified feature of transitional epithelial cells that is mediated by MAPK/ERK activation. We find that two transcription factors, encoded by *Sox2* and *Gata4*, specify primitive transitional epithelium into squamous and columnar epithelium while modulating the diagonal relationship between epithelial *Fgfr2* in the proximal stomach and mesenchymal *Fgf10* in the distal stomach. Thus, the proximal–distal segregation of *Gata4* and *Sox2* expression levels confines the MAPK/ERK-activated transitional epithelial cells within the SCJ boundary during development.

## Results

**Confinement of the primitive KRT7$^+$ transitional epithelium within SCJ during stomach development.** In mouse stomach, the SCJ consists of precisely positioned KRT7$^+$ transitional epithelium between P63$^+$KRT14$^+$LOR$^+$ squamous epithelium in the proximal stomach and GATA4$^+$CLDN18$^+$ columnar epithelium in the distal stomach (Fig. 1a and Supplementary Fig. 1a). At embryonic day (E)18.5, the KRT7$^+$ transitional epithelium can be further divided into KRT7$^+$P63$^+$KRT14$^+$ transitional epithelium at the proximal side and KRT7$^+$P63$^-$KRT14$^-$ transitional

epithelium at the distal side (Fig. 1a). To identify the development of KRT7$^+$ transitional epithelium, we first observed the boundary between P63$^+$ and GATA4$^+$ stomach epithelium, where the SCJ is ultimately formed. Both P63$^+$ proximal stomach epithelium and GATA4$^+$ distal stomach epithelium at E13.5 shape pseudostratified structures with expressing KRT7 (Fig. 1b), histologically similar to KRT7$^+$KRT14$^-$ transitional epithelium at E18.5. In the proximal stomach, the KRT7$^+$ epithelium lacks expression of P63 at E11.5, but starts to express P63 by E13.5 and KRT14 by E15.5 (Fig. 1b and Supplementary Fig. 1b, c). As P63 is required for the formation of a KRT14$^+$ basal layer[12,13], KRT7$^+$P63$^-$KRT14$^-$ epithelium is the primitive epithelial type of the proximal stomach. In the distal stomach, KRT7$^+$ epithelium prominently expressed GATA4 at E11.5 and E13.5 (Fig. 1b). By contrast, the distal stomach epithelium was covered by the KRT7$^-$GATA4$^+$CLDN18$^+$ columnar epithelium at E15.5 and E18.5 (Fig. 1a and Supplementary Fig. 1c), suggesting that KRT7$^+$GATA4$^+$ epithelium differentiate into columnar epithelium. Accordingly, KRT7$^+$ transitional epithelium might be the primitive epithelial type harboring bidirectional differentiation potential into squamous and columnar epithelium in the developing stomach.

**Expression patterning of SOX2 and GATA4 in the stomach epithelium during development.** *Sox2* and *Gata4*, which encode transcription factors, are co-expressed in a stomach primordium at E9.5[15,19]. Both SOX2 and GATA4 are important for the boundary formation of a gastro-intestinal tract; SOX2 defines the boundary of the prospective stomach and intestine around E11.5 in conjunction with CDX2[20], while GATA4 regulates the formation of the posterior boundary between jejunum and ileum in combination with GATA6[21]. It should be noted that SOX2 is predominantly expressed in the squamous epithelium, whereas GATA4 expression is restricted to the columnar epithelium in the newborn mouse[15,22], raising the possibility that SOX2 and GATA4 are involved in the formation of the SCJ boundary. Hence, we examined the expression patterning of SOX2 and GATA4 in the stomach epithelium during development. For that purpose, we performed immunohistochemistry (IHC) together with lineage tracing experiments using *Sox2-CreERT2/+; Rosa^LacZ* mice and *Gata4^CreERT2/+; Rosa^LacZ* mice (Supplementary Fig. 2a, b). SOX2 was broadly expressed in the epithelial cells of the foregut from esophagus to pancreas at E8.5, and SOX2 expression was excluded from pancreas and duodenum by E11.5 (Supplementary Fig. 1d). SOX2 expression gradually formed a proximal–distal gradient in the stomach epithelium from E11.5 to E13.5 and was largely downregulated in the distal stomach at E18.5 (Fig. 1c). By contrast, GATA4 expression shaped the distal–proximal gradient from E9.5 to E11.5 and was subsequently restricted to the distal stomach after E15.5 (Fig. 1c, d). Notably, lineage tracing experiments using *Gata4^CreERT2/+; Rosa^LacZ* embryos showed that GATA4-expressing cells retained the differentiation potential into P63$^+$ basal cells at earlier stages (E9.5–E13.5) but not at E15.5 (Supplementary Fig. 1e).

**SOX2 specifies primitive KRT7$^+$ transitional epithelium into squamous epithelium.** To investigate the roles of SOX2 and GATA4 in the specification of the primitive KRT7$^+$ epithelium, we genetically depleted *Sox2* and *Gata4* in the stomach epithelium during development. We first crossed the *Sox2^CreERT2/+; Rosa^LacZ* mice with *Sox2^flox/flox* mice to obtain *Sox2^CreERT2/flox; Rosa^LacZ* mice, in which SOX2 can be depleted in the stomach epithelium and Cre-mediated recombination can be visualized upon tamoxifen (TAM) treatment. Pregnant females were treated with TAM at E11.5 or E13.5, and the stomach was analyzed at E18.5 (Fig. 2a). Genetic depletion of *Sox2* at E11.5 or E13.5 resulted in the defect of LOR$^+$ keratinized layers and the

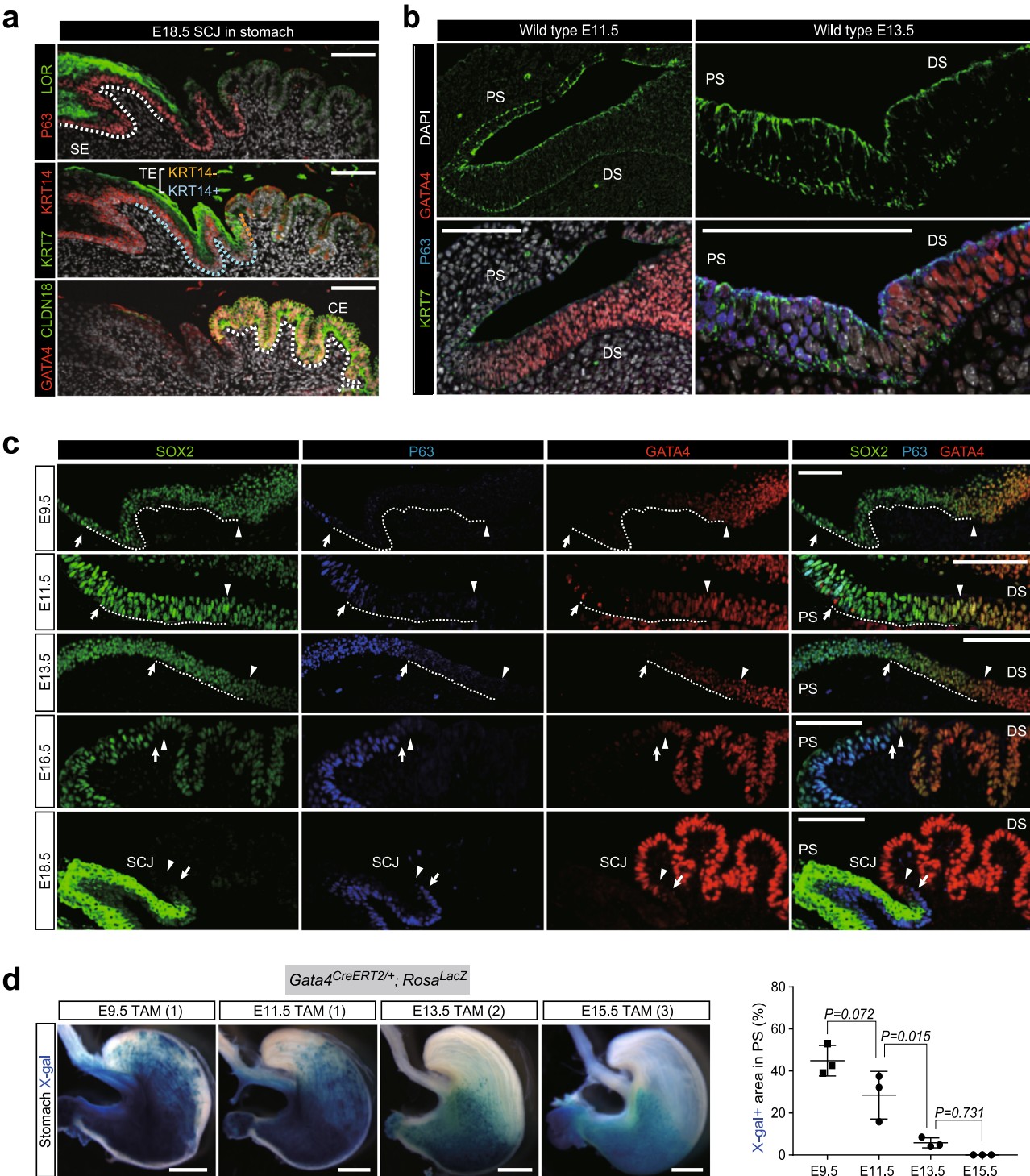

**Fig. 1 Expression patterning of SOX2 and GATA4 in the stomach epithelium during SCJ development. a** Immunofluorescence (IF) analyses of P63 and LOR (top), KRT7 and KRT14 (middle), or GATA4 and CLDN18 (bottom) for the SCJs of wild-type stomachs at E18.5. All samples were counterstained with DAPI. SE squamous epithelium, TE transitional epithelium, CE columnar epithelium. Presented data are a representative image of $n = 5$. Scale bar, 100 μm. **b** IF analyses of KRT7, P63, and GATA4 for wild-type stomachs at E11.5 and E13.5. PS proximal stomach epithelium, DS distal stomach epithelium. Presented data are a representative image of $n = 5$. Scale bar, 100 μm. **c** IF analyses of SOX2, P63, and GATA4 for wild-type stomachs at E9.5, E11.5, E13.5, E16.5, and E18.5. Arrows indicate the P63+ cells on the epithelium at the most caudal side. Arrowheads indicate GATA4+ epithelial cells at the most rostral side. Dashed lines indicate the intermediate epithelial cells with low expression of both P63 and GATA4. Presented data are a representative image of $n = 4$. Scale bar, 100 μm. **d** Left: whole-mount X-gal staining on the stomach of the $Gata4^{CreERT2/+}$; $Rosa^{lacZ}$ embryos. Presented data are a representative image of $n > 8$ embryos out of three independent experiments. Scale bars, 1 mm. Right: quantification of the X-gal+ area in the proximal stomach. Data are presented as mean values ± SD. $n = 3$ independent experiments, one-way ANOVA.

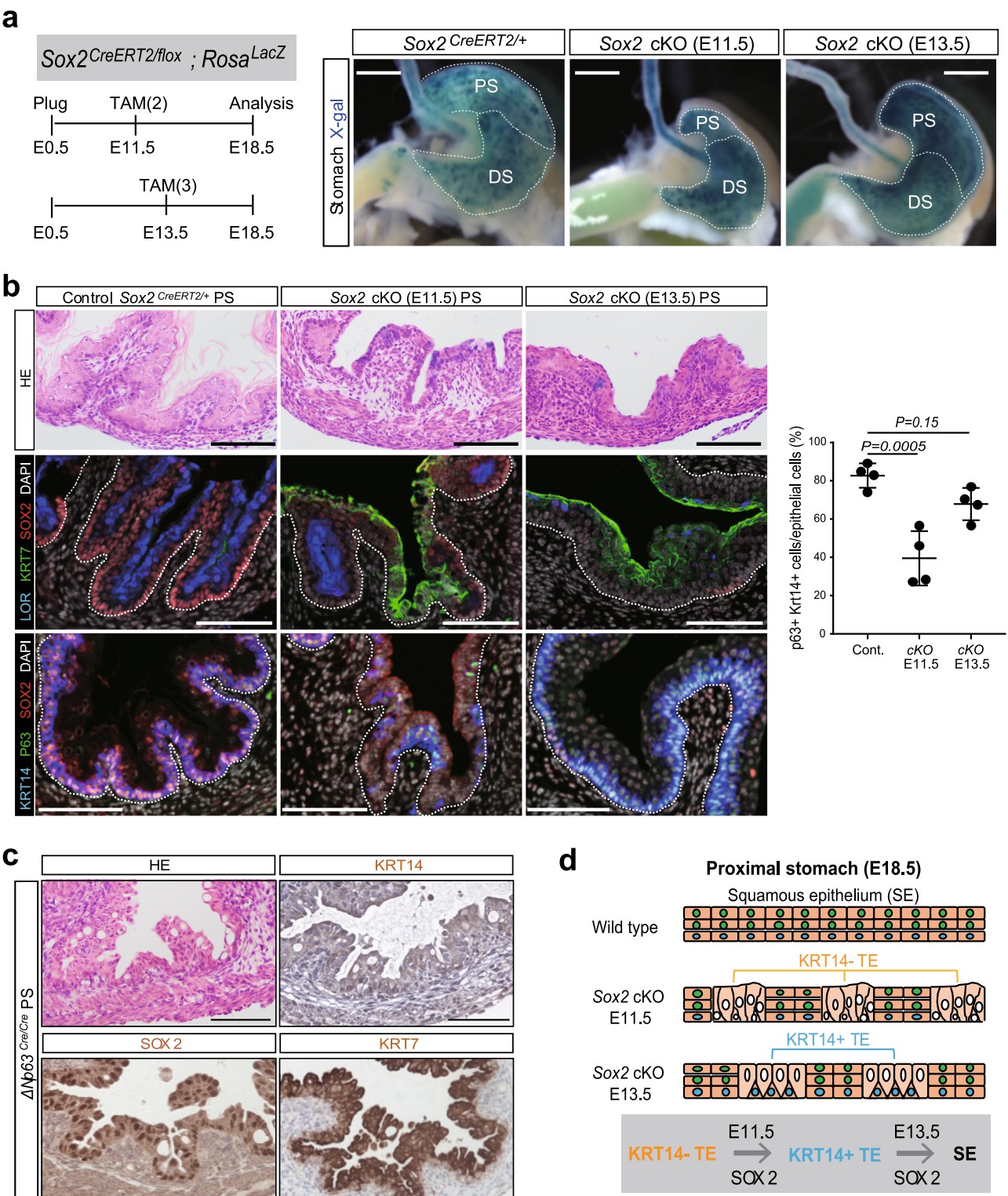

**Fig. 2 SOX2 specifies primitive transitional epithelium into squamous epithelium. a** Left: the scheme for the *Sox2* conditional knock out (*Sox2 cKO*) experiment in the embryonic stomach. Right: the whole-mount X-gal staining on the control, *Sox2 cKO (E11.5)*, and *Sox2 cKO (E13.5)* stomachs at E18.5. Presented data are a representative image of n = 4 embryos out of three independent experiments. Scale bar, 1 mm. **b** Left: H&E staining and IF analyses of LOR, KRT7, and SOX2 (top), or P63, KRT14, and SOX2 (bottom) for the control, *Sox2 cKO (E11.5)*, and *Sox2 cKO (E13.5)* stomachs at E18.5. Presented data are a representative image of n = 4. Scale bar, 100 μm. Right: quantification of the number of the P63$^+$KRT14$^+$ basal cells in the proximal epithelial cells. Data are presented as mean values ± SD. n = 4 independent experiments, one-way ANOVA. **c** H&E staining and IHC analyses of SOX2, KRT14, and KRT7 for the *ΔNp63 KO* proximal stomach at E18.5. Presented data are a representative image of three independent experiments. Scale bar, 100 μm. **d** A scheme illustrates the *Sox2*-mediated specification of the primitive transitional epithelium into the squamous epithelium. SE squamous epithelium, TE transitional epithelium.

replacement of squamous epithelium by KRT7[+] transitional epithelium in the proximal stomachs at E18.5 (Fig. 2b). Close inspections identified that *Sox2* depletion at E11.5 caused the partial expansion of the KRT7[+]KRT14[−] transitional epithelium in the proximal stomach, whereas depletion at E13.5 caused the widespread expansion of KRT7[+]KRT14[+] transitional epithelium (Fig. 2b and Supplementary Fig. 3a–c).

To elucidate the roles of *Sox2* in the differentiation of the primitive transitional epithelium into squamous epithelium, we genetically ablated P63 (specifically the ΔNp63 isoform), which is the master transcription factor for the formation of the stratified squamous epithelium[23,24]. In the ΔNP63-deficient stomach at E18.5, the KRT7[+]KRT14[−] transitional epithelium was persistent in the proximal stomach (Fig. 2c and Supplementary Fig. 3e). ΔNP63-deficient KRT7[+] transitional epithelium expressed SOX2 (Fig. 2c and Supplementary Fig. 3d), indicating that P63 is dispensable for the maintenance of SOX2 expression. Notably, *Sox2* depletion at E11.5 caused the defect of P63 expression (Fig. 2b and Supplementary Fig. 3a), whereas *Sox2* depletion at E13.5 did not affect P63 expression (Fig. 2b), indicating that SOX2 is required for P63 induction but not for the maintenance of P63 expression. Together, our findings demonstrate that SOX2 plays a crucial role in the specification of the primitive KRT7[+]P63[−]KRT14[−] transitional epithelium into KRT7[+]KRT14[+] transitional epithelium, which eventually gives rise to squamous epithelium (Fig. 2d). Even though SOX2 was also expressed in the distal stomach at E11.5, the expression levels of GATA4 and CLDN18 in the distal stomach at E18.5 were not affected by *Sox2* depletion at E11.5 (Supplementary Fig. 3f), implying that SOX2 does not have an impact on the specification of the primitive KRT7[+] transitional epithelial cells into columnar epithelial cells.

**GATA4 specifies primitive KRT7[+] transitional epithelium into columnar epithelium.** We next crossed *Sox2^{CreERT2/+}; Gata4^{flox/flox}; Rosa^{LacZ}* males with *Gata4^{tdTomato/+}* (Supplementary Fig. 2c) females to obtain *Sox2^{CreERT2/+}; Gata4 ^{tdTomato /flox}; Rosa^{lacZ}* mice, in which *Gata4* can be conditionally depleted in the stomach epithelial cells upon TAM treatment, and *Gata4* expression as well as Cre-mediated recombination can be visualized. We treated the pregnant females with TAM at E9.5, and then the stomachs were analyzed at E18.5 (Fig. 3a). Genetic depletion of *Gata4* in the stomach epithelial cells at E9.5 resulted in disruption of pit structures, characteristics of the mature glandular stomach (Fig. 3b). Histologically, GATA4-deficient epithelial cells at the distal stomach lost the columnar epithelial structure and CLDN18 expression, but exhibited a pseudostratified structure with mis-expression of KRT7 (Fig. 3c, d), suggesting that GATA4 specifies the primitive KRT7[+] transitional epithelium into the columnar epithelium. Notably, some ectopic transitional epithelium contained a KRT14[+] basal layer (Fig. 3e). Furthermore, GATA4-deficient KRT7[+] cells in the distal stomach mis-expressed SOX2, and the P63[+]KRT14[+] basal layer was preferentially observed in ectopic KRT7[+] transitional epithelium that mis-expressed SOX2 (Fig. 3e and Supplementary Fig. 4a), supporting our conclusion that SOX2 specifies primitive KRT7[+] transitional epithelium into KRT7[+]KRT14[+] transitional epithelium. Ectopic KRT7[+]KRT14[+] transitional epithelium at E18.5 was also detected following *Gata4* depletion at E11.5, but was almost absent when *Gata4* was depleted at E13.5 (Supplementary Fig. 4b, c). These findings imply that the proximal–distal patterning of the primitive KRT7[+] transitional epithelium is completed around E13.5.

To examine the mechanisms underlying the proximal–distal patterning of the primitive KRT7[+] transitional epithelium mediated by GATA4 and SOX2 expression levels, we performed a gain-of-function experiment. We injected *Col::tetO-Gata4-HA-*

*IRES-mCherry; Rosa^{rtTA}* embryonic stem cells (ESCs) into wild-type blastocysts to generate chimeric mice. We treated pregnant females carrying chimeric embryos with doxycycline (DOX) starting at E11.5 and analyzed the embryos at E18.5 (Fig. 3f). The contribution of *Gata4*-overexpressing cells was visualized by *mCherry* fluorescence (Supplementary Fig. 4d). Overexpression of GATA4 resulted in ectopic emergence of CLDN18[+] columnar epithelium-like cells in the proximal stomach (Fig. 3f). GATA4-overexpressing cells in the proximal stomach epithelium, marked by hemagglutinin (HA) expression, downregulated the expression of SOX2 and its downstream of P63, KRT14, and LOR (Fig. 3g and Supplementary Fig. 4e). Taken together, these observations indicate that GATA4 specifies the primitive KRT7[+] transitional epithelium into the columnar epithelium with downregulating SOX2 expression, which promotes the specification of primitive KRT7[+] transitional epithelium into squamous epithelium (Fig. 3h). In summary, primitive KRT7[+] transitional epithelium is the unspecified epithelial type that is specified into columnar or squamous epithelium, dependent on the expression levels of GATA4 and SOX2.

**Isolation and characterization of the primitive KRT7[+] transitional epithelial cells with different expression levels of *Sox2* and *Gata4*.** Next, we asked how the KRT7[+] transitional epithelium remains unspecified between committed squamous and columnar epithelium in the SCJ. We generated dual-reporter mice (SGGT mice) harboring *EGFP* and *td Tomato* reporter alleles under the control of the *Sox2* and *Gata4* promoters by crossing *Sox2^{EGFP/+}* knock-in mice with *Gata4^{tdTomato/+}* knock-in mice (Supplementary Fig. 2c). We confirmed the results of IHC analyses showing that SOX2 is equally expressed in the proximal and distal stomach at E11.5, but downregulated in the distal stomach at E18.5 (Fig. 4a). GATA4 was broadly expressed in the whole stomach at E11.5, but downregulated in the proximal stomach after E13.5 (Fig. 4a), consistent with the results of the lineage tracing experiment. Using fluorescence-activated cell sorting (FACS), we fractionated EpCAM[+] epithelial cells in the stomach at E11.5, E13.5, and E18.5 into three populations according to their fluorescence intensities (gating strategies are described in Supplementary Fig. 5a), and designated the cells in the three populations as *Sox2*[hi] (*Gata4*[lo]), *Sox2*[mid]*Gata4*[mid], and (*Sox2*[lo]) *Gata4*[hi] (Fig. 4b).

To examine the molecular characteristics of *Sox2*[hi], *Sox2*[mid]*Gata4*[mid], and *Gata4*[hi] cells, we next performed RNA-sequencing (RNA-seq) analyses. We confirmed that *SGGT* fluorescence intensities reflected the expression levels of *Sox2* and *Gata4* (Fig. 4c). Remarkably, *Krt7* was equally expressed in between *Sox2*[hi], *Sox2*[mid]*Gata4*[mid], and *Gata4*[hi] cells, while the expression levels of *Krt14* and *Cldn18* were lower than *Krt7* at E13.5, when the stomach epithelium uniformly assumes the pseudostratified structures but is almost specified into squamous or columnar epithelium (Figs. 1b, d and 4c). Those cellular characteristics were also supported by our RNA-seq results that *Sox2*[hi] cells highly expressed key transcription factors involved in the development of the squamous epithelium, including *p63* and *Foxa2*[18] (Fig. 4c), whereas *Gata4*[hi] cells expressed transcription factors associated with the development of the columnar epithelium, including *Gata6*[21], *Pdx1*[25], and *Hnf4a*[26] (Fig. 4c). Notably, *Sox2*[mid]*Gata4*[mid] cells expressed a negligible level of *p63* relative to *Sox2*[hi] cells, as well as lower levels of *Gata4, Gata6, and Hnf4a* relative to *Gata4*[hi] cells (Fig. 4c), explicitly indicating their unspecified character. In addition, we found that the expression levels of *Cckar, Wnt7b*, and *Syne1* were highest in *Sox2*[mid]*Gata4*[mid] cells (Fig. 4d). RNA in situ hybridization (RNA-ISH) analyses revealed that cells expressing *CCKAR, Wnt7b*, and *Syne1* were localized in the boundary between the GATA4[−] proximal stomach and the GATA4[+] distal stomach at E13.5 and E15.5 (Fig. 4d and

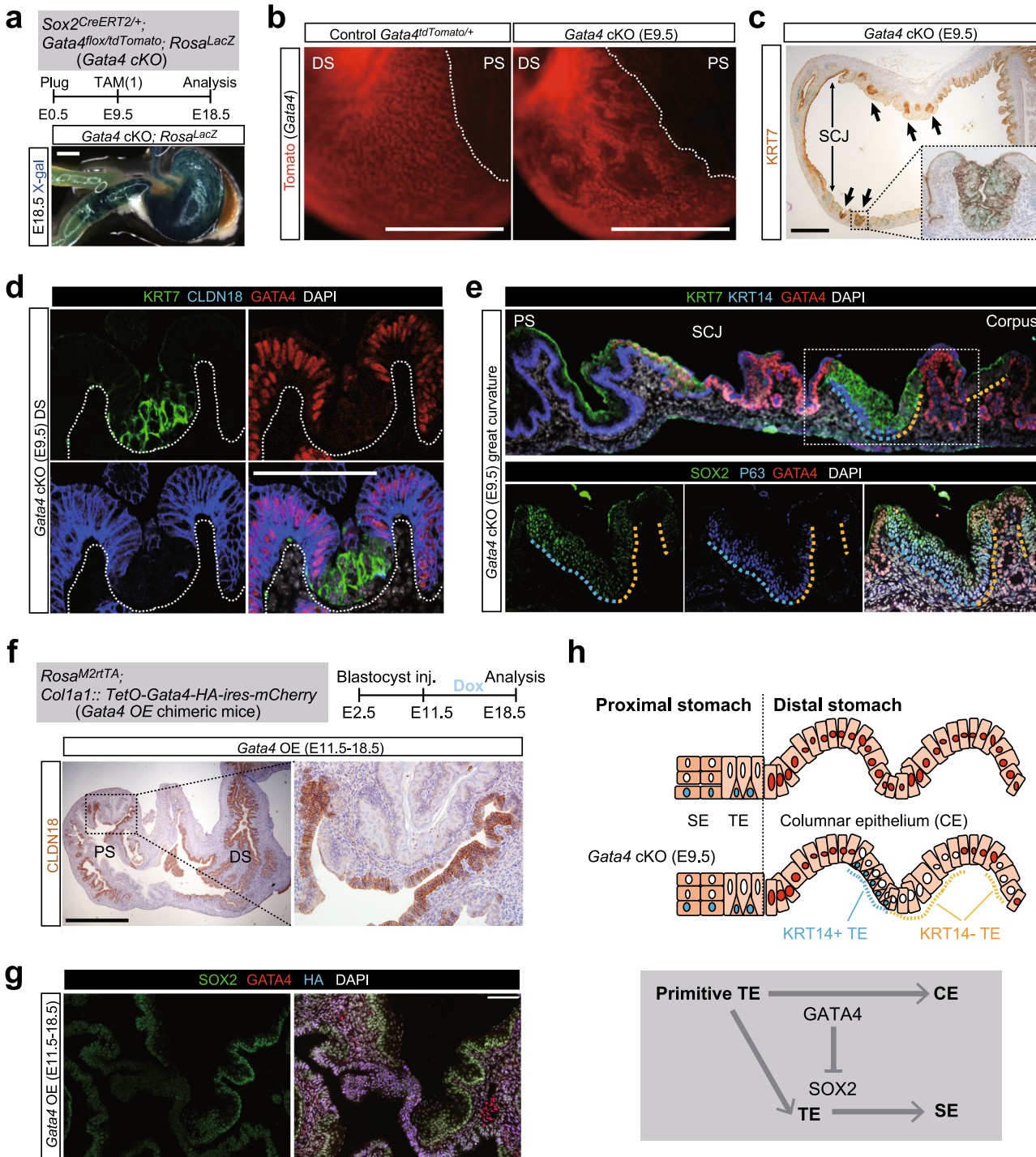

Supplementary Fig. 5b), implying that $Sox2^{mid}Gata4^{mid}$ cells have the unique characteristics and localizes within the boundary of the developing SCJ. Together, our transcriptome analyses suggested that $Sox2^{mid}Gata4^{mid}$ cells at E13.5 may be the unspecified KRT7$^+$ transitional epithelial cells. We also found that the proportion of the $Sox2^{mid}Gata4^{mid}$ cells significantly decreased from E11.5 to E18.5 (28% at E11.5, 19% at E13.5, and 7% at E18.5) (Fig. 4b), presumably because the unspecified KRT7$^+$ transitional epithelial cells are confined to the SCJ boundary during development.

**Sorted $Sox2^{mid}Gata4^{mid}$ cells give rise to both squamous and columnar epithelium in organoids.** To evaluate the differentiation capacities of $Sox2^{hi}$, $Sox2^{mid}Gata4^{mid}$, and $Gata4^{hi}$ cells, we performed organoid experiments. We isolated $Sox2^{hi}$, $Sox2^{mid}Gata4^{mid}$, and $Gata4^{hi}$ cells from the stomach epithelium at E13.5 using FACS, and then cultured the single cells under three-dimensional culture conditions (Fig. 4e). After 10 days, each of the single cells autonomously formed organoids. These organoids could be subdivided into at least three distinct types according to the morphology and expression levels of $Sox2$ and $Gata4$ (Fig. 4f–h). Predominantly SOX2-expressing organoids with smaller surface areas consisted of P63$^+$KRT14$^+$ basal layers and LOR$^+$ cornified layers, which resembled the squamous epithelium (hereafter, squamous epithelium organoid: SEO).

**Fig. 3 GATA4 specifies primitive transitional epithelium into columnar epithelium. a** Top: the scheme for the *Gata4* conditional knock out (*Gata4 cKO*) experiment in the embryonic stomach. Bottom: whole-mount X-gal staining on the stomach of the *Gata4 cKO (E9.5)* embryo at E18.5. Scale bar, 1 mm. **b** Fluorescent images from the *Gata4-td Tomato* of the control and *Gata4 cKO (E9.5)* stomachs at E18.5. Presented data are a representative image of $n = 5$ embryos out of three independent experiments. Scale bar, 1 mm. **c** IHC analyses of KRT7 for the stomachs of the *Gata4 cKO (E9.5)* embryo at E18.5. Arrows indicate the ectopic KRT7$^+$ transitional epithelium in the distal stomach. Presented data are a representative image of $n = 5$ embryos out of three independent experiments. Scale bar, 500 μm. **d** IF analyses of KRT7, CLDN18, and GATA4 for the distal stomach of the *Gata4 cKO (E9.5)* embryo at E18.5. Presented data are a representative image of $n = 5$ embryos out of three independent experiments. Scale bars, 100 μm. **e** IF analyses of KRT7, KRT14, and GATA4 (top), or SOX2, P63, and GATA4 (bottom) for the stomach of the *Gata4 cKO (E9.5)* embryo at E18.5. Presented data are a representative image of $n = 5$ embryos out of three independent experiments. Scale bars, 100 μm. **f** Top: the scheme for the *Gata4* overexpression (*Gata4 OE*) experiment in the embryonic stomach. Bottom: IHC analysis of CLDN18 for the *Gata4 OE* stomach at E18.5. Presented data are a representative image of $n = 4$ chimeric embryos out of three independent experiments. Scale bars, 500 μm. **g** IF analyses of SOX2, GATA4, and HA for the proximal stomach of the *Gata4 OE* embryo at E18.5. Presented data are a representative image of $n = 4$ chimeric embryos out of three independent experiments. Scale bars, 100 μm. **h** A scheme illustrates the *Gata4*-mediated specification of the primitive transitional epithelium into the columnar epithelium. TE transitional epithelium, CE columnar epithelium.

Another type of predominantly SOX2-expressing organoids had larger surface areas that were composed of KRT7$^+$ layers and KRT14$^+$ basal layers, which resembled KRT7$^+$KRT14$^+$ transitional epithelium (hereafter, transitional epithelium organoid: TEO) (Fig. 4g). Predominantly Gata4-expressing organoids consisted of GATA4$^+$CLDN18$^+$ columnar cells, which resembled columnar epithelium (hereafter, columnar epithelium organoid: CEO). CEOs were further subdivided into PDX1$^-$ corpus-type organoids with larger surface areas and PDX1$^+$ antrum-type organoids with smaller surface areas (Fig. 4h). Notably, *Sox2*$^{hi}$ cells mostly gave rise to SEOs, whereas *Gata4*$^{hi}$ cells formed CEOs and *Sox2*$^{mid}$*Gata4*$^{mid}$ cells formed all types of organoids (Fig. 4i). These findings demonstrate that *Sox2*$^{mid}$*Gata4*$^{mid}$ cells have the unspecified properties with multi-lineage differentiation potential in organoids.

**Activation of MAPK/ERK residualizes KRT7$^+$ transitional epithelial cells within SCJ boundary during development**. We next tested the effects of signaling molecules on the cellular differentiation in the organoid experiment and found that treatment with FGF10 significantly changes the morphology and sizes of organoids generated from *Sox2*$^{hi}$ and *Sox2*$^{mid}$*Gata4*$^{mid}$ cells, but not from *Gata4*$^{hi}$ cells (Fig. 5a, b). Importantly, the percentage of TEOs significantly increased when *Sox2*$^{hi}$ cells were cultured with FGF10 (9% and 88% with no cytokine and FGF10 (100 ng/ml), respectively) (Fig. 5c). Consistent with this, the expression of *Krt14* and *Lor* was significantly decreased by FGF10 treatment (Fig. 5d). SEOs were also less abundant in FGF10-treated *Sox2*$^{mid}$*Gata4*$^{mid}$ cells (Fig. 5e). To determine whether the increase in the abundance of TEOs was the result of the FGF10-mediated activation of the intrinsic signaling pathway, we examined the expression of phospho-ERK (pERK), an effector of MAPK/ERK, in each type of organoid. We found that pERK is predominantly expressed in the TEOs (Fig. 5f and Supplementary Fig. 5c). These findings suggest that MAPK/ERK activation blocks the differentiation of transitional epithelial cells into the squamous epithelium.

To assess the physiological activation of MAPK/ERK during development, we next performed IHC analyses of pERK in stomach epithelium. pERK expression was restricted to the KRT7$^+$ transitional epithelium in the SCJ at E18.5 and postnatal day 56 (Fig. 5g). pERK was broadly expressed in the proximal stomach epithelium at E11.5 and E13.5 (Fig. 5h). Subsequently, pERK expression was gradually excluded from the proximal side, and eventually restricted to the SCJ boundary at E18.5 (Fig. 5h). Together, these findings suggest that MAPK/ERK activation residualizes KRT7$^+$ transitional epithelial cells within the SCJ boundary during development.

**Expression patterning of *Fgf10* and *Fgfr2* in the stomach during development**. To examine which signal activates MAPK/ERK

in vivo, we reevaluated our RNA-seq data to search the receptor that potentially activate MAPK/ERK during the SCJ development. Among the representative receptor tyrosine kinases[27], only *Fgfr2* was robustly expressed in the epithelial cells at 11.5 and E13.5 (Supplementary Fig. 6a). We also found that the expression level of *Fgfr2* shapes the gradient, being highest in *Sox2*$^{hi}$ cells, moderate in *Sox2*$^{mid}$*Gata4*$^{mid}$ cells, and lowest in *Gata4*$^{hi}$ cells, a pattern opposite to that of *Gata4* expression (Supplementary Fig. 6a). RNA-ISH analyses revealed that *Fgfr2* expression in the proximal stomach epithelium was weak at E11.5, but became prominent at E13.5 and E15.5 (Fig. 6a). We next searched for FGF ligands expressed in the embryonic stomach by isolating PDGFRα-positive mesenchymal cells at E13.5, E15.5, and E17.5 (Supplementary Fig. 6b). We found that only *Fgf10* was highly expressed in the distal stomach mesenchyme at E13.5 and the *Fgf10* expression in the distal stomach mesenchyme decreased by E17.5 (Supplementary Fig. 6c). Notably, RNA-ISH analyses revealed that *Fgf10* expression in the distal mesenchyme was weak at E11.5, prominent at E13.5, and restricted to the distal end at E15.5 (Fig. 6a). Thus, *Fgf10/Fgfr2* signaling axis in the stomach is established after E11.5, and the distance between *Fgfr2*-expressing epithelial cells and *Fgf10*-expressing mesenchymal cells increased from E13.5 to E15.5 (Fig. 6a). Based on these findings, we propose that the diagonal relationship between epithelial *Fgfr2* and mesenchymal *Fgf10*, along with their spatiotemporal patterning, is responsible for the regionally restricted activation of MAPK/ERK during the establishment of the SCJ in the stomach.

**Epithelial expression levels *of Gata4* and *Sox2* regulate intermediated regional activation of MAPK/ERK**. We next examined whether the proximal–distal patterning of *Sox2/Gata4* expression interacts with MAPK/ERK activation mediated by the *Fgfr2/Fgf10* axis. First, we sought to determine why *Gata4*$^{hi}$ cells did not respond to *Fgf10* in the organoid experiment (Fig. 5a, b), even though *Fgf10* was released from the neighboring mesenchymal cells and *Fgfr2* was partially expressed in *Gata4*$^{hi}$ cells (Supplementary Fig. 6a, c). From our RNA-seq data, we discovered that genes related to the negative regulation of the MAPK/ERK cascade, such as *Spry2*, *Dusp4*, and *Dusp6*, were highly expressed in the *Gata4*$^{hi}$ cells relative to *Sox2*$^{hi}$ and *Sox2*$^{mid}$*Gata4*$^{mid}$ cells at E13.5 (Supplementary Fig. 6d). Depletion of *Gata4* in the stomach epithelium at E11.5 using *Sox2*$^{CreERT2/+}$; *Gata4*$^{flox/flox}$ embryos resulted in the mis-expression of pERK in the GATA4-deficient epithelial cells at E13.5 (Fig. 6b), which persisted in the ectopic GATA4-deficient KRT7$^+$ transitional epithelium in the distal stomach at E18.5 (Fig. 6c), demonstrating that the negative regulation of MAPK/ERK in the stomach epithelium is dependent on epithelial GATA4 expression. Furthermore, when we blocked MAPK/ERK signaling by treating E11.5 embryonic stomach with SU5402 in ex vivo

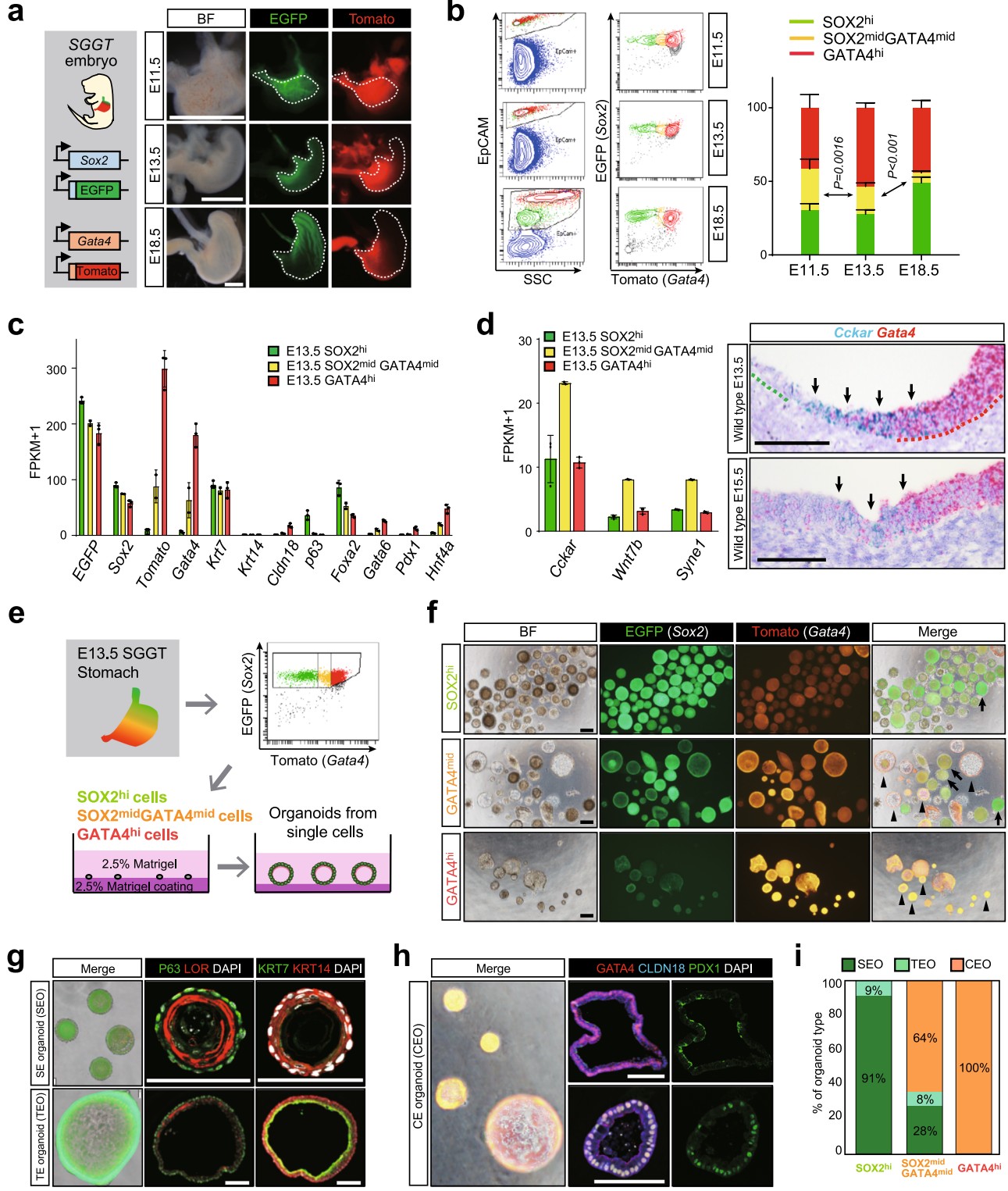

culture, the number of *Gata4*hi cells in the stomach epithelium decreased (Supplementary Fig. 6e). This implies the existence of a negative-feedback loop between epithelial GATA4 and MAPK/ERK activation. By contrast, when we almost completely deleted *Gata4* in the stomach epithelium from E11.5 to E13.5 using *Gata4*CreERT2/flox embryos (Fig. 6d), the activation of MAPK/ERK in the proximal stomach epithelium was significantly downregulated (pERK+/epithelial cells = 25% [control] vs. 7% [Gata4 cKO]) (Fig. 6h). Notably, expression of *Fgf10* in the distal mesenchyme was downregulated in

*Gata4*-depleted stomach at E13.5 (Fig. 6e), indicating that epithelial GATA4 expression regulates the mesenchymal *Fgf10* expression.

We next examined the roles of epithelial *Sox2* in MAPK/ERK activation. Pregnant females carrying *Sox2*CreERT2/flox embryos were treated with TAM at E11.5, and their stomachs were analyzed at E13.5 (Fig. 6f). The number of pERK+ epithelial cells in the proximal stomach decreased in *Sox2*-depleted stomachs (pERK+/epithelial cells = 25% [control] vs. 5% [*Sox2* cKO]) (Fig. 6h). Notably, expression of *Fgfr2* was downregulated in

**Fig. 4 *Sox2*<sup>mid</sup>*Gata4*<sup>mid</sup> cells are the unspecified transitional epithelial cells with multi-lineage differentiation potential. a** Left: a scheme for generation of the *Sox2-EGFP, Gata4-td Tomato* dual-reporter mouse (*SGGT*). Right: fluorescence images of *SGGT* stomachs at E11.5, E13.5, and E18.5. Presented data are a representative image of $n > 5$. Scale bar, 1 mm. **b** Left: FACS analyses for *SGGT* stomachs at E11.5, E13.5, and E18.5 using EpCAM staining. Right: quantification of the percentages of *Sox2*<sup>hi</sup>, *Sox2*<sup>mid</sup>*Gata4*<sup>mid</sup>, and *Gata4*<sup>hi</sup> cells in stomach epithelial cells at E11.5 ($n = 10$), E13.5 ($n = 8$), and E18.5 ($n = 5$). Data are presented as mean values ± SD, one-way ANOVA. **c** RNA-seq analyses for *Sox2*<sup>hi</sup> ($n = 3$), *Sox2*<sup>mid</sup>*Gata4*<sup>mid</sup> ($n = 2$), and *Gata4*<sup>hi</sup> ($n = 3$) cells of embryonic stomach at E13.5. Data are presented as mean values ± SD. **d** Left: RNA-seq analyses of *Cckar*, *Wnt7b*, and *Syne1* for *Sox2*<sup>hi</sup> ($n = 3$), *Sox2*<sup>mid</sup>*Gata4*<sup>mid</sup> ($n = 2$), and *Gata4*<sup>hi</sup> ($n = 3$) cells of embryonic stomach at E13.5. Data are presented as mean values ± SD. Right: RNA in situ hybridization (RNA-ISH) analyses of *Cckar* for wild-type stomach at E13.5 (top) and E15.5 (bottom). Arrows indicate *Cckar*<sup>+</sup> cells located in intermediate epithelium. Presented data are a representative image of $n = 3$ embryos. Scale bars, 100 µm. **e** A scheme for organoid experiments using single *Sox2*<sup>hi</sup>, *Sox2*<sup>mid</sup>*Gata4*<sup>mid</sup>, and *Gata4*<sup>hi</sup> cells isolated from *SGGT* stomach at E13.5 by FACS. **f** Fluorescence images of organoids generated from the *Sox2*<sup>hi</sup>, *Sox2*<sup>mid</sup>*Gata4*<sup>mid</sup>, and *Gata4*<sup>hi</sup> cells. Arrows indicate squamous epithelium organoids (SEOs), and arrowheads indicate columnar epithelium organoids (CEOs). $n = 4$. Scale bars, 100 µm. **g** Fluorescence images of SEOs and transitional epithelium organoids (TEOs) (left) and IF analyses of P63 and LOR (middle) or KRT7 and KRT14 (right) for SEOs and TEOs. $n = 5$. Scale bars, 100 µm. **h** Fluorescence images of CEOs (left) and IF analyses of GATA4, CLDN18, and PDX1 for CEOs. $n = 5$. Scale bars, 100 µm. **i** Quantification of organoid types generated from *Sox2*<sup>hi</sup>, *Sox2*<sup>mid</sup>*Gata4*<sup>mid</sup>, and *Gata4*<sup>hi</sup> cells. Data represent averages of four independent experiments.

Sox2-deficient epithelial cells in the proximal stomach (Fig. 6g), suggesting that epithelial SOX2 expression regulates the epithelial *Fgfr2* expression. To confirm this idea, we performed experiments of GATA4 overexpression from E11.5 to E13.5 using *KH2-Gata4* chimeric embryos, in which GATA4-overexpressing cells in the proximal stomach epithelium downregulated the expression levels of SOX2 and its target genes (Fig. 3g and Supplementary Fig. 4e). We found that expression levels of SOX2 and *Fgfr2* were decreased in GATA4-overexpressing cells in the proximal stomach epithelium at E13.5. In conclusion, epithelial expression levels of *Gata4* and *Sox2* orchestrates the intermediate regional activation of MAPK/ERK mediated by *Fgf10/Fgfr2* axis during development (Fig. 6i).

**Activation of MAPK/ERK in human transitional epithelium and Barrett's esophagus.** Finally, we examined the activation of MAPK/ERK signaling at SCJs in humans. Different from mouse, the SCJ is not located in the stomach but at the esophageal–gastric junction in human[12,13]. However, the human SCJs have common structures with the mouse SCJs in that KRT7<sup>+</sup> transitional epithelial cells exist between squamous and columnar epithelium[13,28]. We detected specific expression of pERK in KRT7<sup>+</sup> transitional epithelium in the human esophageal–gastric junction, as well as KRT7<sup>+</sup> transitional epithelium in the uterine cervix (Fig. 7a). Thus, the activation of MAPK/ERK cascade is a general feature of KRT7<sup>+</sup> transitional epithelium at the SCJs in both mice and humans.

Recent studies suggest that transitional basal cells are an origin of the Barrett's esophagus, an intestinal metaplasia that arises specifically at the esophageal–gastric junction[12,13]. Barrett's esophagus is defined as the ectopic emergence of goblet cells around the SCJ and harbors a higher risk for adenocarcinoma development[4]. Transcriptome analyses comparing Barrett's metaplasia with squamous and columnar epithelium demonstrated that Barrett's metaplasia expresses high levels of *KRT7* and *FGFR2* (Fig. 7b)[17]. By contrast, expression of *SOX2* remained suppressed, whereas expression of GATA4 was upregulated in Barrett's metaplasia (Fig. 7b), suggesting that Barrett's metaplasia exhibits a transcription pattern similar to that of primitive transitional epithelial cells. Expression of pERK was detected in a subset of cells in Barrett's metaplasia (Fig. 7c). Moreover, IHC analyses of Ki67 revealed that the pERK<sup>+</sup> cells in Barrett's metaplasia were highly proliferative (Fig. 7c). Together, these findings support the notion that primitive transitional epithelial cells and activation of MAPK/ERK signaling are associated with the pathogenesis of Barrett's metaplasia.

## Discussion

In this study, we identified the molecular mechanisms underlying the establishment of the SCJ boundary. We found that during development, pERK-positive primitive epithelial cells are confined to transitional epithelium at the SCJ boundary. Confinement of the pERK-positive cells is governed by the coordinated interplay between a concentration gradient of a signaling molecule (*Fgf10–Fgfr2* axis) and patterning of transcription factors (*Gata4–Sox2*). Moreover, transcription factor-mediated cell fate specification in epithelial cells affects mesenchymal expression of signaling molecules that play roles in the activation of the MAPK/ERK cascade.

Previous studies using tissue-swapping or explant culture experiments showed that mesenchyme directs epithelial cell fate in the stomach[29,30]. Additionally, it was reported mesenchymal *Fgf10* affects the cellular differentiation of the gastric gland in mice[14,31], supporting the notion that mesenchymal signals direct the epithelial cell fate. Conversely, we showed that the epithelial deletion of *Gata4* caused mesenchymal depletion of *Fgf10* expression in distal stomach, highlighting the local crosstalk between epithelial cells and mesenchymal cells. We also found that *Sox2* mediates epithelial expression of *Fgfr2* in the proximal stomach. Notably, the distance between *Fgfr2*-expressing epithelial cells and *Fgf10*-expressing mesenchymal cells increased along with the growth of the stomach during development. Collectively, these results suggest that distant epithelial–mesenchymal crosstalk straddling the SCJ ensures the regional specificity of MAPK/ERK activation, leading to the persistence of pERK-positive transitional epithelial cells at the SCJ.

Given their unique localization, transitional epithelial cells have been implicated in tissue regeneration and maintaining homeostasis of SCJs. Here, by studying the SCJ development in mouse stomach, we showed that the activation of MAPK/ERK mediated by *Fgf10–Fgfr2* axis maintains the proliferative and unspecified properties of embryonic TE cells, which have the potential to differentiate into both squamous and columnar epithelial cells. Notably, activation of MAPK/EPK in the transitional epithelial cells was preserved after the establishment of the SCJ. Since the expression of *Fgf10* in the mesenchyme decreased at the later developmental stage, it remains to be clear how MAPK/ERK is activated in adult transitional epithelial cells. Nevertheless, these results support the hypothesis that pERK-positive transitional epithelial cells might serve as undifferentiated stem cells capable of regenerating and maintaining the homeostasis of the adult SCJ. Future studies, using lineage tracing of the pERK-positive transitional epithelial cells and manipulating MAPK/ERK activation in the adult transitional epithelial cells, might verify this hypothesis.

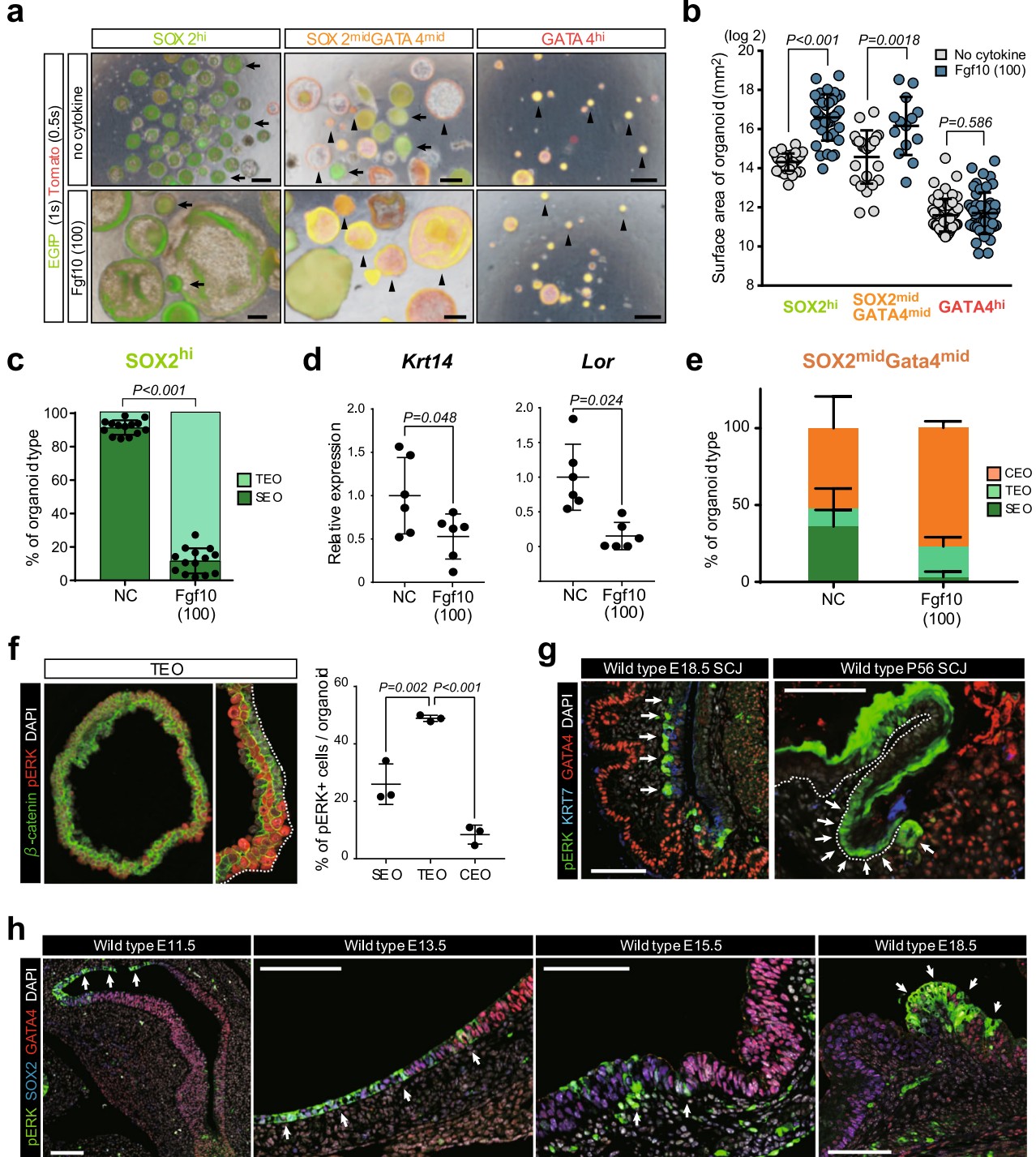

**Fig. 5 Activation of MAPK/ERK prevents the differentiation of KRT7+ transitional epithelial cells. a** Fluorescent images of organoids generated from *Sox2*[hi], *Sox2*[mid]*Gata4*[mid], and *Gata4*[hi] cells in the culture condition with no cytokine (NC) or FGF10 (100 ng/mL). Arrows indicate the SEOs and arrowheads indicate CEOs. *n* = 5. Scale bar, 100 μm. **b** Surface areas of organoids generated from *Sox2*[hi], *Sox2*[mid]*Gata4*[mid], and *Gata4*[hi] in the culture condition with NC or FGF10 (100 ng/mL). *n* > 13 organoids of three independent experiments, two-sided *t*-test. **c** Quantification of organoid types generated from *Sox2*[hi] cells in the culture condition with NC or FGF10 (100 ng/mL). *n* = 14 independent experiments, two-sided *t*-test. **d** Quantitative (Q)-PCR analyses of *Krt14* and *Lor* for organoids generated from *Sox2*[hi] cells in the culture condition with NC or FGF10 (100 ng/mL). The CT value of each gene is normalized by *B2M*. The average ΔCT value of the NC condition is set to 1. *n* = 6 independent experiments. Data are presented as mean values ± SD, two-sided *t*-test.

**e** Quantification of organoid types generated from *Sox2*[mid]*Gata4*[mid] cells in the culture condition with no cytokine (NC) or FGF10 (100 ng/mL). *n* = 4 independent experiments. Data are presented as mean values ± SD. **f** Left: IF analysis of phospho-ERK (pERK) and β-catenin for TEOs. Presented data are a representative image of *n* > 10 organoids out of three independent experiments. Scale bar, 100 μm. Right: quantification of pERK+ cells in SEOs, TEOs, and CEOs. *n* = 3 independent experiments, one-way ANOVA. **g** IF analyses of pERK, KRT7, and GATA4 for the wild-type stomachs at E18.5 and postnatal day 56. Arrows indicate the pERK+KRT7+ transitional epithelial cells in the SCJ. Presented data are a representative image of *n* = 5. Scale bar, 100 μm. **h** IF analyses of pERK, SOX2 and GATA4 for wild-type stomachs at E11.5, E13.5, E15.5, and E18.5. Arrows indicate the pERK+ epithelial cells in the boundary between proximal and distal stomachs. Presented data are a representative image of *n* = 5. Scale bar, 100 μm.

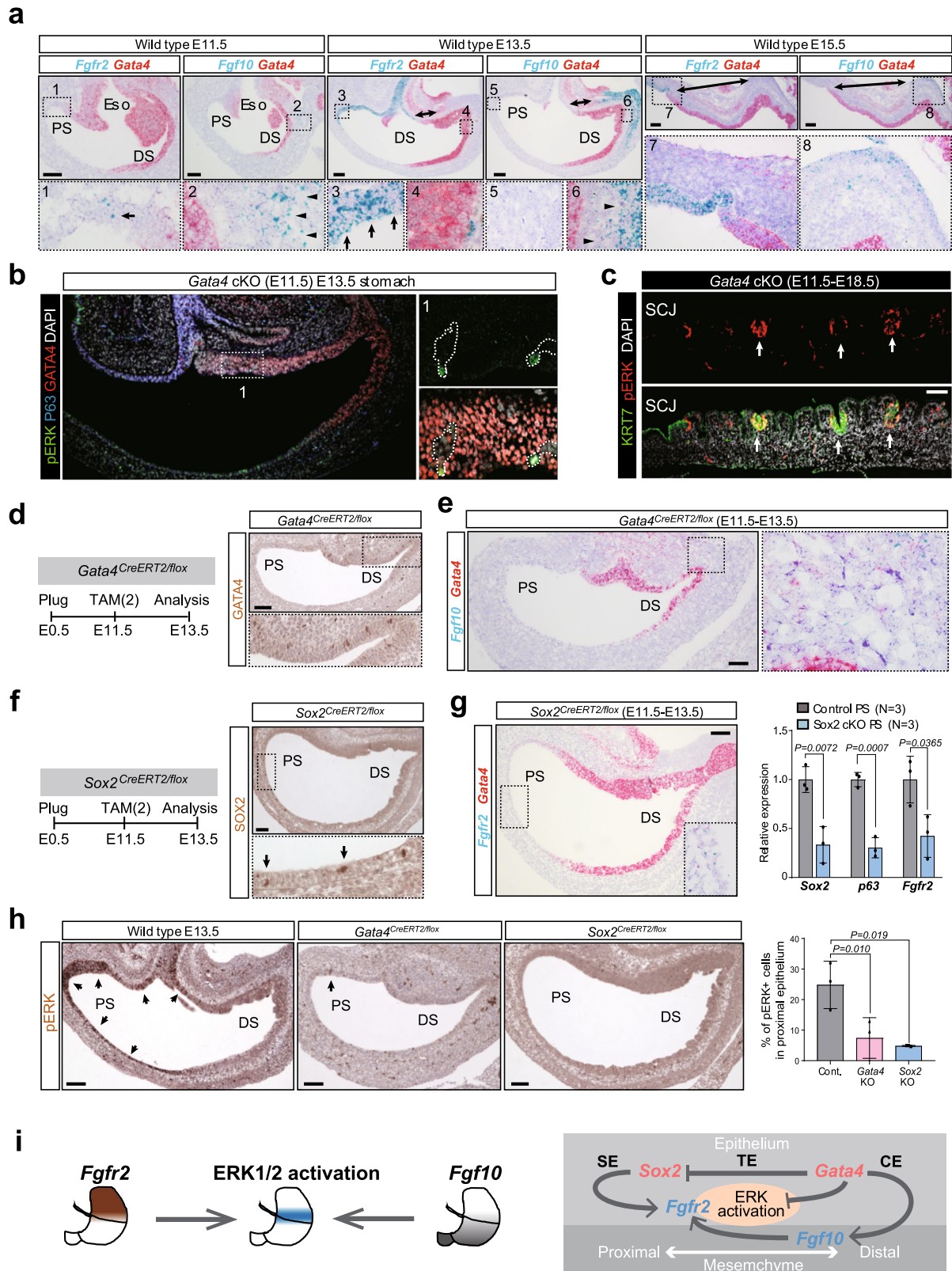

Notably, FGF10-mediated activation of MAPK/ERK is efficient for the long-term expansion of human Barrett's metaplasia in organoids[32]. Furthermore, activation of MAPK/ERK is frequently observed in adenocarcinomas associated with Barrett's metaplasia[7,33]. We found that a subset of proliferating cells in Barrett's metaplasia expresses pERK, suggesting that these pERK-positive cells could be precursors of Barrett's metaplasia-associated adenocarcinomas. Therefore, activation of MAPK/ERK signaling might play an important role not only in the pathogenesis of Barrett's metaplasia, but also in development of esophageal adenocarcinomas from

**Fig. 6 Epithelial expression levels of *Gata4* and *Sox2* regulate the regionally restricted activation of MAPK/ERK via *Fgfr10/Fgfr2* axis. a** RNA-ISH analyses of *Fgf10*, *Fgfr2*, and *Gata4* for wild-type stomachs at E11.5, E13.5, and E15.5. Arrows indicate the *Fgfr2*-expressing proximal epithelial cells. Arrowheads indicate the *Fgf10*-expressing distal mesenchymal cells. *n* = 4. Scale bar, 100 μm. **b** IF analyses of pERK, P63, and GATA4 for the stomachs of the *Gata4* cKO (E11.5) embryos at E13.5. Dash lines demarcated the *Gata4*-depleted cells. *n* = 3. Scale bar, 100 μm. **c** IF analyses of pERK and KRT7 for the stomach of the *Gata4* cKO (E11.5) embryo at E18.5. Arrows indicate the pERK⁺KRT7⁺ transitional epithelial cells in the distal stomach. *n* = 3. Scale bar, 100 μm. **d** Left: a scheme for the *Gata4* KO experiments using *Gata4^CreERT2/flox^* mice. Right: IHC analyses of GATA4 for the stomachs of the *Gata4^CreERT2/flox^* embryos at E13.5. *n* = 3. Scale bar, 100 μm. **e** RNA-ISH analyses of *Fgf10* and *Gata4* for the stomachs of the *Gata4^CreERT2/flox^* embryos at E13.5. *n* = 3. Scale bar, 100 μm. **f** Left: a scheme for the *Sox2* cKO experiment using *Sox2^CreERT2/flox^* mice. Right: IHC analyses of SOX2 for the stomachs of the *Sox2^CreERT2/flox^* embryos at E13.5. *n* = 3. Scale bar, 100 μm. **g** Left: RNA-ISH analyses of *Fgfr2* and *Gata4* for the *Sox2* cKO (E11.5) stomachs at E13.5. *n* = 3. Scale bar, 100 μm. Right: Q-PCR analyses of *Sox2*, *p63*, and *Fgfr2* for the epithelial cells of control and *Sox2* cKO (E11.5) stomachs at E13.5. The CT value of each gene is normalized by *B2M*. The average ΔCT values of control stomachs are set to 1. *n* = 3 independent experiments. Data are presented as mean values ± SD, two-sided *t*-test. **h** Left: IHC analyses of pERK for the stomachs of wild type, *Gata4^CreERT2/flox^*, and *Sox2^CreERT2/flox^* embryos at E13.5. Arrows indicate the pERK⁺ cells in the stomach epithelium. Scale bar, 100 μm. Right: quantification of pERK⁺ cells in the proximal stomach epithelium. *n* = 3 independent experiments. Data are presented as mean values ± SD, one-way ANOVA. **i** A scheme illustrates that epithelial *Gata4* and *Sox2* regulate the axial patterning of *Fgf10/Fgfr2* and regionally restricted activation of MAPK/ERK.

---

Barrett's metaplasia, and thus may represent a potential therapeutic target.

In summary, we identified the unspecified features of transitional epithelium in the SCJ, which is maintained by continuous activation of MAPK/ERK. Our findings pave the way to an improved understanding of pathogenesis and will facilitate the development of strategies for cancer treatment, as well as efficient regeneration, both of which are related to SCJs.

## Methods

### Vector constructions

**Gata4-td Tomato *and* Gata4-CreERT2 *vector*.** A cDNA fragment of *td Tomato-pA-PGK-Puro-pA* (2.7 kb) and *CreERT2-pA-PGK-Bsd-pA* (3.8kbp) with 50-bp homology arms was established using KAPA HiFi HotStart ReadyMix (KAPA Biosystems). This fragment was recombined into the first ATG of the exon 1 of the *Gata4* BAC (BACPAC Resources Center) using the Red/ET BAC recombination system. The fragments of *Gata4-td Tomato-pA-PGK-Puro-pA* and *Gata4-CreERT2-pA-PGK-Bsd-pA* sequence with 2.5-kb (5′) and 4.2-kb (3′) homology arms were retrieved and used as targeting vectors.

**Sox2-CreERT2 *vector*.** A cDNA fragment of *CreERT2-pA-PGK-Bsd-pA* (3.8 kb) with 50-bp homology arms was established using KAPA HiFi HotStart ReadyMix. This fragment was recombined into the first ATG of the exon 1 of the *Sox2* BAC (BACPAC Resources Center) using the Red/ET BAC recombination system. The fragment of *Sox2-CreERT2-pA-PGK-Bsd-pA* sequence with 3.5-kb (5′) and 2.5-kb (3′) homology arms was retrieved and used as a targeting vector.

**Homologous recombination into ES cells.** ES cell lines V6.5 and KH2 were used as previously described[34,35]. 20 μg of each targeting vector was linearized using the appropriate restriction enzymes (Gata4-td Tomato targeting vector: ScaI, LSL-HA tag-Kras^G12D^ targeting vector: PvuI, Gata4-CreERT2 targeting vector: FsbI, Sox2-CreERT2 targeting vector: SbfI). All restriction enzymes were obtained from New England Biolabs. After 37 °C overnight linearization, the products were purified and collected by ethanol precipitation; the pellets were dissolved in 100 μL of 25 mM HEPES buffer (Gibco). ES cells were dissociated with 0.25% trypsin (Nacalai Tesque) and resuspended in 500 μL of high-glucose DMEM (Nacalai Tesque) containing 25 mM HEPES. A mixture of targeting vector and ES cells (4.0 × 10⁶) was injected into a Gene Pulser Cuvette (Bio-Rad) and subjected to electroporation on a Gene Pulser Xcell (Bio-Rad). Electroporation was performed twice at 550 V for 600 msec. After electroporation, 2.0 × 10⁶ ES cells were placed onto MEFs in each of two 6-cm dishes and cultured at 37 °C with 5% CO₂ in ESC medium [Knockout DMEM (Gibco), 2 mM L-glutamine (Nacalai Tesque), 1× NEAA (Nacalai Tesque), 100 U/mL penicillin, 100 μg/mL streptomycin (Nacalai Tesque), 15% FBS (Gibco), 0.11 mM mercaptoethanol (Gibco), and 1000 U/mL human LIF (Wako)]. Twenty-four hours after electroporation, antibiotic selection was initiated. Concentrations were as follows: blasticidin S (Funakoshi), 15 μg/mL and puromycin (Sigma), 1 μg/mL. After 1 week of antibiotic selection, the surviving ES colonies were picked up and expanded to establish ESC lines.

**Southern blotting.** Genomic DNA was collected from ES cells using the NucleoSpin Tissue kit (Macherey-Nagel). The recombined BAC plasmids were collected using the NucleoBond Xtra BAC kit (Macherey-Nagel). 3 ng of each recombined BAC (as a positive control) and 5 μg of sample genomic DNA were digested overnight at 37 °C with the appropriate restriction enzymes (Gata4-td Tomato: KpnI, Gata4-CreERT2: HindIII, Sox2-CreERT2: MfeI). Restriction

enzymes were obtained from New England Biolabs. The restriction products were ethanol precipitated and dissolved in 15 μL of TE buffer. The products were mixed with 5 mL of 6× Loading Buffer Triple Dye (NIPPON Gene) and subjected to electrophoresis on a 0.8% agarose gel (Fast Gene) at 50 V for 2 h. The products in the gel were depurinated in 250 mM HCl for 10 min, washed in distilled water for 5 min, and denatured in 0.5 M NaOH, 1.5 M NaCl for 15 min; the last step was repeated twice. Subsequently, the products were transferred to a Hybridization Transfer Membrane (PerkinElmer) by capillary transfer at room temperature for 8 h. The membrane was washed twice with 2× Standard Saline Citrate (SSC buffer) for 5 min and crosslinked on a CL-1000 Ultraviolet Crosslinker (UVP). 20× SSC buffer contains 3 M of NaCl, 0.3 M of C₆H₅Na₃O₇•2H₂O. The pH of the samples was adjusted to 7.0 using HCl. The membrane was hybridized at 68 °C overnight with 25 ng/mL probe in PerfectHyb buffer (TOYOBO). Probes were generated using the PCR DIG Probe Synthesis Kit (Roche) using the wild-type BAC as a template. The primer sets for each target gene are shown in Supplementary Table 1. The membrane was washed twice with low-stringency buffer (containing 2 × SSC buffer and 1 × SDS), and twice with high-stringency buffer (containing 0.2 × SSC buffer, 1 × SDS); all washes were for 5 min. The DIG Luminescent Detection Kit for Nucleic Acids (Roche) was used for visualization, and an LAS4000 (GE Healthcare) was used for detection. Uncropped blots are found in Supplementary Fig. 7.

**Blastocyst collection and microinjection**

*Blastocysts collection.* Eight-week-old ICR female mice (Japan SLC) received 7.5 U of serotropin (ASKA Animal Health) by intraperitoneal injection. Forty-eight hours after of serotropin treatment, mice were injected with 7.5 U of gonadotropin (ASKA Pharmaceutical). These mice were then mated with ICR male mice (Japan SLC). Plug checks were performed on the next morning. Two days later, these female mice were sacrificed by cervical dislocation, and their oviducts were harvested. Two-cell-stage fertilized eggs were collected by perfusion with M2 medium (Sigma) and maintained in KSOM medium. Two days later, the blastocysts were subjected to microinjection. For this purpose, ES cells were treated with trypsin and pipetted up and down 15 times to dissociate them into single cells. The ES cells and MEFs were incubated in a gelatin-coated 10-cm dish with 10 mL ESC medium for 30 min to attach only MEFs onto the dish. 3 mL of supernatant containing ES cells was collected. Three to five ES cells were injected into each ICR blastocyst under on OLYMPUS IX71 microscope. In all, 20–25 injected blastocysts were transplanted into the uterus of each pseudo-pregnant ICR female mouse (Japan SLC).

**Mice.** *Rosa26 LSL-LacZ* mice[36], *ΔNp63-Cre* mice[23], *Sox2-EGFP* mice[37], *Sox2 flox* mice[38], and *Gata4 flox* mice[39] have been described previously. All mice were maintained in a C57BL/6–129×1/Sv mixed background. For germline transmission, 8-week-old male chimeric mice were mated with 8-week-old C57BL/6 female mice (Japan SLC) to obtain transgenic mice.

**Mice genotyping.** Tail tips of 3-week-old mice were collected and dissolved with 200 μL of DNA elution buffer (100 mM Tris-HCl, 5 mM EDTA, 0.2 % SDS, 200 mM NaCl, and 1% Protein kinase) at 65 °C. After centrifugation at 4 °C at 20,000 × g for 15 min, 3 μL of supernatant was dissolved into 100 μL of Tris-EDTA buffer. Recombination was detected by PCR with Gflex polymerase (Takara Bio). Genotyping primers are shown in Supplementary Table 1.

**Tamoxifen and doxycycline treatment.** TAM (Sigma) was dissolved in corn oil. For embryonic labeling and genetic manipulation, pregnant females were received a single-dose intra-abdominal injection of TAM at the times indicated. DOX (Sigma) was dissolved in drinking water at 0.2 mg/mL and was given to pregnant females for the times indicated.

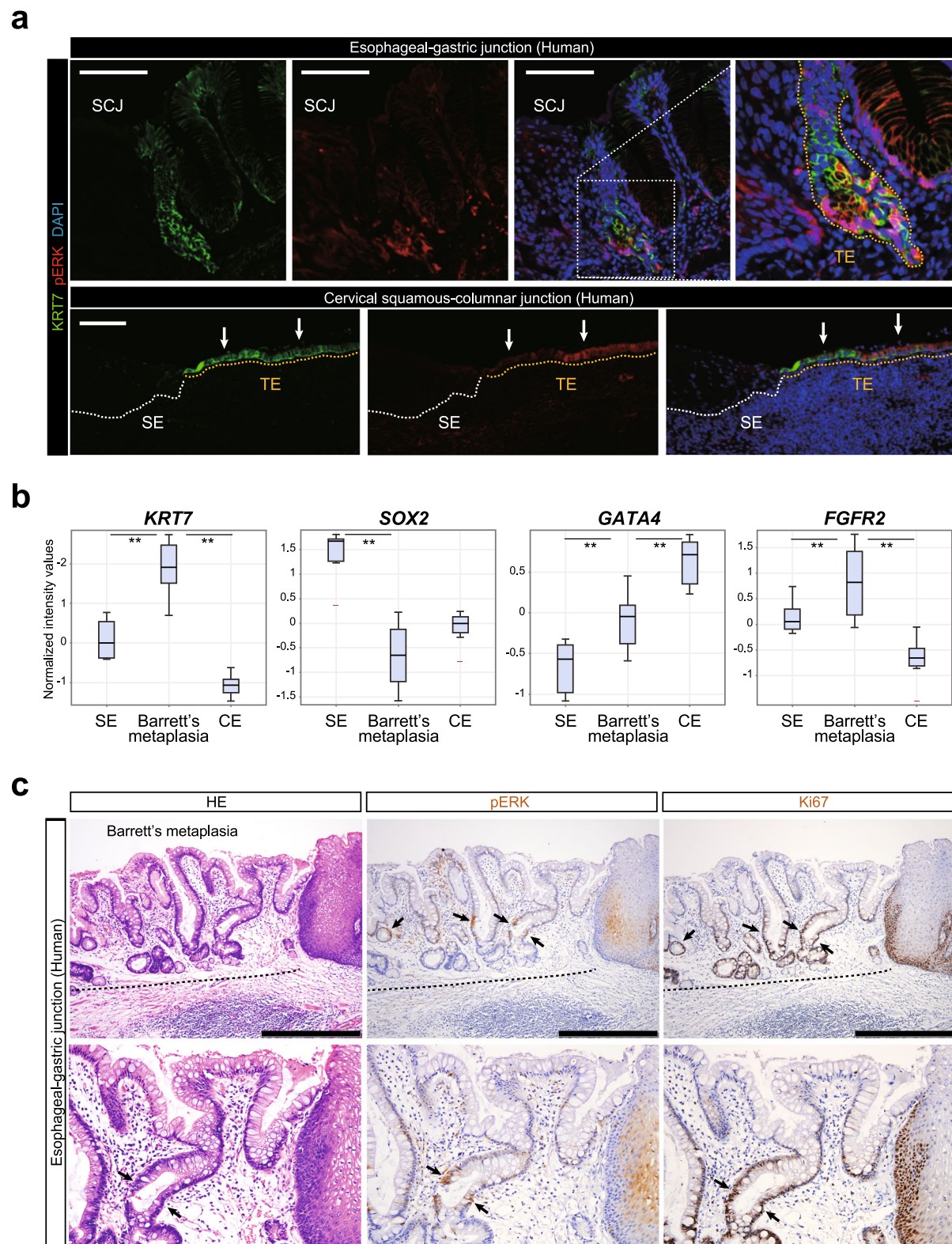

**Fig. 7 Activation of MAPK/ERK in human SCJs and Barrett's esophagus. a** IF analyses of pERK and KRT7 for human esophageal–gastric junction and uterine cervix. Arrows indicate the pERK+KRT7+ transitional epithelial cells. SE squamous epithelium, TE transitional epithelium. Presented data are a representative image of $n = 4$. Scale bar, 100 μm. **b** Expression levels analyses of *KRT7*, *SOX2*, *GATA4*, and *FGFR2* for esophageal squamous epithelium (SE) ($n = 8$), gastric columnar epithelium (CE) ($n = 10$), and Barrett's metaplasia ($n = 10$). Raw data are extracted from GEO record GSE34619. Solid lines in each box indicate the median. Bottom and top of the box are lower and upper quartiles, respectively. **$p < 0.01$, one-way ANOVA. **c** H&E staining (left) and IHC analyses of pERK (middle) and Ki67 (right) in human Barrett's metaplasia. Arrows indicate pERK+Ki67+ cells. Top: ×10 magnifications. Bottom: ×20 magnifications. Presented data are a representative image of $n = 4$. Scale bar, 500 μm.

**Mice experiments**. All animal studies were conducted in accordance with ethical guidelines in Kyoto University and University of Tokyo and were approved by Center for iPS Cell Research and Application, Kyoto University and Institute of Medical Science, University of Tokyo.

**X-gal staining**

*Whole-mount staining*. Pregnant females were sacrificed, and the dissected intestinal tracts of *Gata4-CreERT2, Rosa LSL-LacZ* and *Sox2-CreERT2, Rosa LSL-LacZ* embryos were collected and fixed with ice-cold 4% PFA for 2 h. After one wash with PBS, the stomach was poured into permeabilization solution (5 mM EGTA, 2 mM $MgCl_2$, 0.01% sodium deoxycholate, and 0.02% Nonidet P-40 in PBS) and reacted with the X-gal solution (1 mg/mL X-gal, 5 mM potassium ferrocyanide and 5 mM potassium ferricyanide in the permeabilization solution) at 4 °C for overnight.

*Frozen tissue staining*. After fixation of dissected tissues with PBS containing 30% sucrose, the tissues were embedded in Tissue-Tek O.C.T compound (Sakura). Frozen tissues were sectioned at 100-µm thickness, sliced serially into 7-µm-thick sections, and stained as described above. Counterstaining was performed using Contrast Red (KPL).

**RNA extraction from tissues and organoids**. The tissues and organoids were collected and dipped into 1 mL of Sepazol (Nacalai Tesque), shattered with a sonicator (Qsonica) for 30 s, and mixed with 200 µL of chloroform (Wako). After shaking for 15 s, the mixtures were left to stand for 2 min at room temperature and centrifuged at 4 °C for 15 min at $12,000 \times g$. 300 µL of supernatants was carefully collected and mixed with 300 µL of 70% ethanol. RNA was extracted using RNeasy spin columns (Qiagen).

**Quantitative RT-PCR**. RNA was extracted using the RNeasy Plus Micro Kit or Mini Kit (Qiagen). RNA was quantified with NanoDrop (Thermo Scientific). 500 ng of RNA was used for the reverse transcription reaction into cDNA using ReverTra Ace (Toyobo). Real-time quantitative PCR was performed using SYBR Green qPCR Master Mix (Roche). Transcript levels were normalized against the corresponding levels of *β2-microglobulin* mRNA. PCR primers are listed in Supplementary Table 1.

**Paraffin-embedded specimen preparation**. Dissected tissue samples were fixed in 4% PFA (Wako) overnight at room temperature. The next day, the samples dehydrated through an ethanol series: 70% ethanol for several hours, followed by 80, 90, and 100% ethanol and Histo-Clear (Wako). After dehydration, the samples were embedded in paraffin using a spin tissue processor (STP120, Thermo Scientific). Sections were sliced into 4-µm-thick sections and stained with hematoxylin and eosin (Muto Pure Chemicals); serial sections were used for immunohistochemical analyses.

**Immunostaining**. Samples were soaked in xylene twice for 5 min to remove paraffin, and then 100% ethanol twice 3 min to hydrophilize. After washing with water in several minutes, the samples were soaked into epitope retrieval buffer (DAKO) and autoclaved at 105 °C for 10 min. The samples were then soaked into 1× PBS for several minutes and incubated with 100 µL of primary antibody in IMMUNO SHOT (Cosmo Bio) at 4 °C overnight. Antibodies and dilutions are shown in Supplementary Table 2. After two 5-min washes with PBS, the samples were incubated for 30 min at room temperature with secondary antibodies. The samples were washed twice in PBS for 5 min, and then nuclear staining was performed with hematoxylin for 10–30 s. After washing with water in 10 min, samples were placed in 100% ethanol (three times), and then placed in xylene three times. Finally, the samples were evaluated under a microscope (Keyence).

**Immunofluorescence staining**. The procedure was as described above for immunostaining until the primary antibody treatment. The samples were processed with DAPI (1:500) (Invitrogen) and fluorescently labeled secondary antibodies (1:500) diluted in PBS, and incubated for 60 min in room temperature. Antibodies and dilutions are shown in Supplementary Table 2. After two washes in PBS for 5 min, the samples were mounted and evaluated by confocal laser scanning microscopy (Zeiss LSM700).

**FACS**. Stomachs from E11.5 and E13.5 *Sox2-EGFP/Gata4-td Tomato* embryos were dissected and treated with Accumax at 37 °C for 30 min. Stomachs from E18.5 *Sox2-EGFP/Gata4-td Tomato* embryos were dissected and treated with 0.25% Trypsin for more than 30 min. After centrifugation ($300 \times g$ for 3 min), dispersed cells were rinsed with PBS, and then stained with the Alexa Fluor 647-conjugated EpCAM antibody on ice for 30 min. Dead cells were gated out by DAPI staining.

**RNA sequencing**. Total RNA (50 ng) was prepared for library construction. High-quality RNA (RNA integrity number value ≥ 9), assessed by a Bioanalyzer, was used for the library preparation. RNA-seq libraries were generated using the Truseq Stranded mRNA LT sample prep kit (Illumina). PolyA-containing mRNA was purified using poly-T oligoattached magnetic beads, and the RNA was fragmented and primed for cDNA synthesis. Cleaved RNA fragments were reverse transcribed into first strand cDNA using transcriptase and random primers. Second strand cDNA was synthesized by the incorporation of dUTP, and ds cDNA was separated using AMPure XP beads (BECKMAN COULTER). A single "A" nucleotide was added to the 3′ ends of the blunt fragments, and then the indexed adapters were ligated to the ends of the ds cDNA. ds cDNA fragments were amplified by PCR with a PCR primer Cocktail. The number of PCR cycles was minimized (11–15 cycles) to avoid skewing the representation of the libraries. RNA-seq libraries were sequenced on NextSeq500 (75 bp single, Illumina).

**RNA in situ hybridization**. RNA-ISH was performed on paraffin-embedded tissues fixed in 4% PFA for 24 h following the procedure described above for paraffin-embedded specimen preparation. All probes and RNAscope 2.5HD assay-Duplex were purchased from Advanced Cell Diagnostics, and ISH was performed according to the manufacture's protocol. After sections were pretreated with Pretreats 1–3, probes were hybridized in the HybEZ oven for 2 h at 40 °C. Signals were amplified with AMP1–6, and then with alkaline phosphatase. After further amplification of the signals with AMP7–9, signals were detected by DAB and counterstained with Mayer's hematoxylin. Probes used in this study are listed in Supplementary Table 2. Note that the probe for *Gata4* is designed to bind the undeleted locus of *Gata4* mRNA of *Gata4^CreERT2/flox* embryos that enabled to mark the GATA4-deficient epithelial cells in Fig. 6e.

**Organoid culture**. Epithelial cells were gathered from embryonic stomachs by FACS as described above. Four-well dishes (Nunc) were coated with 4% Matrigel (Corning) in 375 µL RPMI for 60 min, and then sorted cells were replated in Advanced DMEM containing B27, N2, HEPES, Glutamax, P/S, and 2.5% Matrigel with or without Fgf10 (R&D Systems). Medium was changed every 2 days.

**Stomach organ culture**. Stomachs were isolated from the *SGGT* embryos at E11.5 and cultured on filters (Millicell, 0.4 µm; Merk Millipore) in DMEM/F12 containing P/S and 10% FBS with or without SU5402 (Sigma Aldrich).

**Patients derived paraffin-embedded tissue samples**. Patients derived paraffin-embedded tissue samples were used in accordance with ethical guidelines in Kyoto University Hospital. All patients were provided informed consent for use of the tissue samples in this research.

**Statistics and reproducibility**. All data are presented as means ± SD and represent a minimum of three independent experiments. Statistical parameters including statistical analysis, statistical significance, and *n* values are described in the figure legends. Statistical analyses were carried out using Prism 7 Software (GraphPad). Statistical comparisons of two groups were performed using the two-sided unpaired *t*-test, and comparisons of more than three groups were performed by one-way ANOVA. A value of $p < 0.05$ was considered significant. Presented data are a representative image of $n > 3$ out of more than three biologically independent experiments.

**Reporting summary**. Further information on research design is available in the Nature Research Reporting Summary linked to this article.

## Data availability
The authors declare that all data supporting the findings of this study are available within the article and its Supplementary Information files or from the corresponding author upon reasonable request. RNA-seq data reported in this study have been deposited in the Gene Expression Omnibus (GEO) database under the accession code: GSE143217.

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

## Acknowledgements
We thank K. Mitsunaga for help with FACS analyses. We thank M. Kabata and J. Asahira for technical support with RNA-seq analyses. This work was supported by JSPS KAKENHI (Grant Numbers JP26253069 and JP16J01756); Core Center for iPS Cell Research, Research Center Network; Grant for International Joint Research Project of the Institute of Medical Science, University of Tokyo.

## Author contributions
N.S., W.T. and Y.K. conceived the study and designed the experiments, acquired data, performed analyses, and interpreted the results. T.C. and S.U. supervised the experiments. A.T., H.S., K.W., T.Y. and Y.Y. provided technical assistance. A.T., H.S. and Y.Y. assisted in mouse experiments. Y.S. and M.M. provided clinical samples. H.H. and Y.Y contributed to clinical data analyses. K.W. and Y.Y. provided materials. N.S. and Y.K. wrote the paper.

## Competing interests
The authors declare no competing interests.
