## [Peer Review File · Nature Communications]

Reviewers' Comments:

Reviewer #1:

Remarks to the Author:

This manuscript combined mouse genetics, RNA sequencing and organoid culture to address the roles for Sox2 and Gata4 in the specification of the transitional epithelium bridging squamous and columnar epithelium in the proximal-distal stomach boundary. The major findings include the alternation of the transitional epithelium following deletion of Sox2 or Gata4. Notably, organoid culture of Sox2midGata4mid epithelial cells isolated from E13.5 stomach demonstrates capability of squamous and columnar differentiation. Moreover, Sox2midGata4mid organoids also showed a significantly higher expression of phospho-ERK by FGF10 treatment compared to Sox2hi and Gata4hi organoids. The authors further show that MAPK/ERK activation has a role in the formation/maintenance of the transitional epithelium. In the human squamous-columnar junction, p-ERK expression is also enriched in the transitional epithelium. Further data show low levels of Sox2 and Gata4 and high levels of KRT7 and Fgfr2 expression in Barrett's esophagus. These findings are interesting and significant, providing important information about the transcription factors and signaling pathways regulating the transitional epithelium which has been shown to play important roles in Barrett's metaplasia and neoplasia in other transitional zones. The manuscript can be further strengthened with following minor comments.

1. Lineage tracing with Gata4-creER mouse line is interesting. Can the authors provide more detailed characterization of lineage labeled cells at the PS, TE and DS, e.g. Xgal-co-staining with H&E, Xgal containing with Krt7, especially for Tmx injected at E13.5 and E15.5?
2. Fig2B. The authors need to show HE staining of the junction following the loss of Sox2 (E11.5 TMX injection). It is not clear for the transitional epithelium with DAPI staining. HE staining will provide more details.
3. Can the authors show HE staining of the SCJ shown in Fig2D? It is difficult to see the details of the epithelium in the mutants.
4. Fig2D. Based on this study and previous findings by Jiang et al., 2017, KRT7+ cells were initially present in the esophagus and proximal stomach when the esophagus/stomach established from the foregut. Therefore, in this reviewer's opinion it will be more appropriate to say "Following deltaNp63 deletion KRT7+ cells are persistent" instead of saying "expanded in the proximal stomach" at E11.5.
5. Can the authors examine Fgfr2 expression in the squamous-columnar junction following p63 deletion?

Reviewer #3:

Remarks to the Author:

The manuscript by Sankoda et al. describes how expression of transcription factors Gata4 and Sox2 may regulate establishment of squamous-columnar junction (SCJ) in the mouse stomach during embryonic development. The authors define cells located within SCJ as KRT7+ P63- KRT14- transitional epithelium (TE) cells. By using organoid system, the authors report that Fgf10 expression mediated by Gata4 in the distal (glandular) stomach and Fgfr2 expression mediated by Sox2 in the proximal (squamous) stomach lead to localized activation of MAPK/ERK in cells of SCJ. Furthermore, based on immunostainings and transcriptome analysis the authors report activation of MAPK/ERK in human TEs and Barrett's esophagus. They suggest that MAPK/ERK signaling is associated with the pathogenesis of Barrett's metaplasia. They also propose that their findings support a possibility that TE cells are the cells of origin of Barrett's metaplasia. The study has several strengths, such as generation of new Cre-deleter mice allowing careful lineage tracing and interrogation of Gata4 and Sox2 expressing populations. Organoid systems are also well applicable to questions raised. However, some conclusions are insufficiently supported by presented data and some issues need to be addressed.

- 1) All mouse studies has been performed on mouse embryos. What does happen to TE cells in adult mice? Are they also regulated by Gata2/FGf10 and Sox2/Fgfr2 signaling and MAPK/ERK activation?
- 2) The offered evidence that KRT7+ P63- KRT14- cells represent bona fide TE cells with a potential for differentiation in vivo is not entirely convincing without KRT7 lineage tracing.
- 3) Are KRT7+ cells essential for formation of columnar and/or squamous epithelium? What does happen after their deletion in SCJ?
- 4) How large is a fraction of Sox2midGata4mid cells expressing with KRT7+ P63- KRT14- phenotype? Perhaps a fluorescent KRT7 reporter would be a better option for studies of TE cells?
- 5) Offering some parallels between the embryonic mouse stomach epithelium to Barrett's esophagus has its origins in the manuscript by Wang et al. in 2011. It is an interesting but very speculative possibility. Without experimentation in adult mice and human organoids, this part of the manuscript is not on very solid ground.

Minor:

- 1) Explanation/justification for the mouse stomach SCJ as a parallel for human gastro-esophageal junction needs to be provided.
- 2) In Fig 1b: At E13.5, KRT7+ cells seems to be present in different areas of stomach epithelium, and not as a single layer described by authors.
- 3) In Extended Fig 1c, KRT7 expression is difficult to see and interpret. It seems to be on the surface of the stratified epithelium and also in stroma.
- 4) Line 99-100: Without KRT7 lineage tracing Fig. 1a, b suggests, not shows, "differentiation" of KRT7+GATA+ cells into KRT7- GATA4+ CLDN18+ 100 columnar epithelium by E18.5.
- 5) Perhaps authors need to provide a more thorough review of recent (and quite plentiful) literature on the SCJ.

RE: NCOMMS-20-20262

Response to referee's comments

Reviewer #1:

We would like to thank the reviewer for his/her time and suggestions on helping us improve our manuscript. Our responses to each point are listed. The reviewers' comments are underlined.

This manuscript combined mouse genetics, RNA sequencing and organoid culture to address the roles for Sox2 and Gata4 in the specification of the transitional epithelium bridging squamous and columnar epithelium in the proximal-distal stomach boundary. The major findings include the alternation of the transitional epithelium following deletion of Sox2 or Gata4. Notably, organoid culture of Sox2midGata4mid epithelial cells isolated from E13.5 stomach demonstrates capability of squamous and columnar differentiation. Moreover, Sox2midGata4mid organoids also showed a significantly higher expression of phospho-ERK by FGF10 treatment compared to Sox2hi and Gata4hi organoids. The authors further show that MAPK/ERK activation has a role in the formation/maintenance of the transitional epithelium. In the human squamous-columnar junction, p-ERK expression is also enriched in the transitional epithelium. Further data show low levels of Sox2 and Gata4 and high levels of KRT7 and Fgfr2 expression in Barrett's esophagus. These findings are interesting and significant, providing important information about the transcription factors and signaling pathways regulating the transitional epithelium which has been shown to play important roles in Barrett's metaplasia and neoplasia in other transitional zones.

We thank the reviewer for his/her positive comments.

The manuscript can be further strengthened with following minor comments.

1) Lineage tracing with Gata4-creER mouse line is interesting. Can the authors provide more detailed characterization of lineage labeled cells at the PS, TE and DS, e.g. Xgal-co-staining with H&E, Xgal containing with Krt7, especially for Tmx injected at E13.5 and E15.5?

Response

According to this reviewer's suggestion, we performed X-gal-co-staining with H&E, KRT7 and P63 for Gata4-CreERT2; Rosa-reporter embryos labeled by Tmx injections at E9.5, E11.5, E13.5 and E15.5 (revised Extended data Fig 1e). We detected X-gal⁺/KRT7⁺ TE cells in the embryos labeled at E9.5, E11.5, and E13.5 but not at E15.5. X-gal⁺/P63⁺ TE basal cells were frequently observed in the proximal stomach labeled at E9.5 and E11.5, very few at E13.5 and none at E15.5. Thus, a subset of GATA4⁺ cells retains a potential of squamous cell differentiation by E13.5 but lost it after E15.5. These results are consistent with our findings that Gata4 expression was detected in the whole stomach at first, formed a gradient pattern from E9.5 to E13.5, and was finally restricted to the distal epithelium around E16.5 (Fig 1c, 4a, 4b).

2) Fig2B. The authors need to show HE staining of the junction following the loss of Sox2 (E11.5 TMX injection). It is not clear for the transitional epithelium with DAPI staining. HE staining will provide more details.

Response

According to the reviewer's suggestion, we showed HE staining for the proximal stomach epithelium of Sox2 cKO (E11.5 and E13.5) embryos (revised Fig 2b). While the control stomach had squamous epithelium with a keratinized layer, the proximal stomach epithelium of Sox2 cKO embryos broadly defected the keratinized layer and the squamous epithelium was replaced by the TE.

3) Can the authors show HE staining of the SCJ shown in Fig2D? It is difficult to see the details of the epithelium in the mutants.

Response

We performed HE staining for the SCJ of the Δ Np63 KO stomach (revised Extended data Fig 3e). The proximal stomach epithelium of the Δ Np63 KO embryo was pseudostratified and morphologically similar to the TE of the control stomach. The distal stomach has the normal columnar epithelium in the Δ Np63 KO embryo.

4) Fig2D. Based on this study and previous findings by Jiang et al., 2017, KRT7+ cells were initially present in the esophagus and proximal stomach when the esophagus/stomach established from the foregut. Therefore, in this reviewer's opinion it will be more appropriate to say "Following deltaNp63 deletion KRT7+

cells are persistent” instead of saying “expanded in the proximal stomach” at E11.5.

Response

We agree with this reviewer’s comment and revised the description as follows:

~ In the $\Delta NP63$ -deficient stomach at E18.5, the primitive $KRT7^+ KRT14^-$ TE was persistent in the proximal stomach epithelium (Fig. 2c, Extended Data Fig. 3e), being consistent with the results of previous studies about P63-deficient stomachs^{12,13}

5) Can the authors examine Fgfr2 expression in the squamous-columnar junction following p63 deletion?

Response

We performed qPCR analyses for the $\Delta Np63^{Cre/Cre}$ stomach at E18.5 and showed that Fgfr2 expression was not significantly downregulated comparing to control $\Delta Np63^{Cre/+}$ proximal stomach (revised Extended Data Fig. 3d). P63 was yet to be expressed in the proximal stomach before E13.5, while Fgfr2 was expressed in the proximal stomach epithelium at E11.5 (Fig. 5A, Extended Data Fig. 6a), suggesting that P63 does not directly regulate Fgfr2 expression.

Reviewer #3:

We would like to thank the reviewer for his/her time and suggestions on helping us improve our manuscript. Our responses to each point are listed. The reviewers’ comments are underlined.

The manuscript by Sankoda et al. describes how expression of transcription factors Gata4 and Sox2 may regulate establishment of squamous-columnar junction (SCJ) in the mouse stomach during embryonic development. The authors define cells located within SCJ as KRT7+ P63- KRT14- transitional epithelium (TE) cells. By using organoid system, the authors report that Fgf10 expression mediated by Gata4 in the distal (glandular) stomach and Fgfr2 expression mediated by Sox2 in the proximal (squamous) stomach lead to localized activation of MAPK/ERK in cells of SCJ. Furthermore, based on immunostainings and transcriptome analysis the authors report activation of MAPK/ERK in human TEs and Barrett’s esophagus. They suggest that MAPK/ERK signaling is associated with the pathogenesis of Barrett’s metaplasia.

They also propose that their findings support a possibility that TE cells are the cells of origin of Barrett's metaplasia. The study has several strengths, such as generation of new Cre-deleter mice allowing careful lineage tracing and interrogation of Gata4 and Sox2 expressing populations. Organoid systems are also well applicable to questions raised.

Response

We thank for the reviewer's positive comments.

However, some conclusions are insufficiently supported by presented data and some issues need to be addressed.

1. All mouse studies has been performed on mouse embryos. What does happen to TE cells in adult mice? Are they also regulated by Gata2/FGf10 and Sox2/Fgfr2 signaling and MAPK/ERK activation?

Response

As the reviewer pointed, this study was mainly focused on revealing the physiological properties of embryonic TE cells. Thus, the characteristics of adult TE cells were out of scope in this study. To address the role of MAPK/ERK activation and Fgf10/Fgfr2 axis on the TE cells after establishment the SCJ, we performed additional RNA-seq analyses of PDGFRa⁺ stomach mesenchymal cells at E17.5. We found that Fgf10 expression in the distal stomach was significantly decreased by E17.5 (revised Extended Data Fig. 6c). By contrast, MAPK/ERK was activated in the TE cells of the established SCJ at E18.5 and in adult stages in adult stage (Fig. 5g, 7a). Based on this finding, we assume that role of Fgf10/Fgfr2 axis on MAPK/ERK activation in TE cells becomes less significant after the establishment of SCJ, and identification of other MAPK/ERK activator(s) at these stages is the focus of the future studies. Accordingly, we revised the manuscript as follows:

~ Notably, activation of MAPK/EPK in the TE cells was preserved after the establishment of the SCJ. As the expression of Fgf10 in the mesenchyme decreased at the later developmental stage, it remains to be clear how MAPK/ERK is activated in adult TE cells. Nevertheless, these results support the hypothesis that pERK-positive TE cells might serve as undifferentiated stem cells capable of regenerating and maintaining the homeostasis of the adult SCJ. Future studies, using lineage tracing of the

pERK-positive TE cells and manipulating MAPK/ERK activation in the adult TE cells, might verify this hypothesis.

2. The offered evidence that KRT7+ P63- KRT14- cells represent bona fide TE cells with a potential for differentiation in vivo is not entirely convincing without KRT7 lineage tracing.

Response

Our RNA-seq and IHC analyses showed that KRT7 was broadly expressed in the epithelium of both proximal stomach and distal stomach at 13.5 and of the proximal stomach epithelium at E15.5 (revised Fig. 1b, 4c, revised Extended Data Fig. 1c), thus KRT7 expression was not localized to the KRT7⁺ P63⁻ KRT14⁻ TE cells before E15.5. Therefore, it is technically difficult to trace the cellular fate of the KRT7⁺ P63⁻ KRT14⁻ cells using KRT7 lineage tracing in vivo. Our RNA-seq analyses suggested that Sox2^{mid}Gata4^{mid} cells at E13.5 were mainly KRT7⁺ P63⁻ KRT14⁻. Therefore, lineage tracing of Gata4 at this stage enabled us, though indirectly, to track the fate of KRT7⁺ P63⁻ KRT14⁻ cells in vivo. Gata4 lineage tracing showed the subsets of Gata4-expressing cells at E13.5 could differentiate into both squamous and columnar epithelial cells in vivo (Fig. 1d, revised Extended data Fig. 1e). In addition, we confirmed the multi-differentiation potential of Sox2^{mid} Gata4^{mid} cells at E13.5 in vitro.

3. Are KRT7+ cells essential for formation of columnar and/or squamous epithelium? What does happen after their deletion in SCJ?

Response

We thank for this reviewer's helpful suggestion. During the development of the SCJ, KRT7 was broadly expressed in the stomach epithelium and the expression of KRT7 was gradually restricted to the boundary between proximal and distal stomach epithelium (revised Fig. 1b, 4c). Therefore, KRT7⁺ cells are indispensable for the formation of all epithelial types constituting SCJ before E13.5. Although the role of the TE cell for the maintenance of the adult SCJ is out of scope in this study, it should be addressed at next time. We discussed these points in the Discussion section as follows; ~ Future studies, using lineage tracing of the pERK-positive TE cells and manipulating MAPK/ERK activation in the adult TE cells, might verify this hypothesis.

4. How large is a fraction of Sox2^{mid}Gata4^{mid} cells expressing with KRT7+ P63- KRT14- phenotype? Perhaps a fluorescent KRT7 reporter would be a better option for studies of TE cells?

Response

Again, during the development of the SCJ, KRT7 is broadly expressed in the stomach epithelium at E13.5 and the expression is gradually restricted to the boundary (Fig. 1b, 4c, Extended Data Fig. 1c). Therefore, all Sox2^{mid}Gata4^{mid} cells at E13.5 express KRT7. Our RNA-seq analyses demonstrated that Sox2^{mid}Gata4^{mid} cells expressed negligible levels of P63 and Krt14 (revised Fig. 4c). Together, we assume that most of Sox2^{mid}Gata4^{mid} cells at E13.5 are KRT7⁺ P63⁻ KRT14⁻. Accordingly, we revised the manuscript as follows: ~ Remarkably, Krt7 was equally expressed in between Sox2^{hi}, Sox2^{mid}Gata4^{mid}, and Gata4^{hi} cells while the expressions of Krt14 and Cldn18 were lower than Krt7 at E13.5, when the stomach epithelium uniformly assumes the TE histology but is almost specified into squamous or columnar epithelium (Fig. 1b, d, 4c). Those cellular characteristics were also supported by our RNA-seq results that Sox2^{hi} cells highly expressed key transcription factors involved in the development of the squamous epithelium, including p63 and Foxa2¹⁸ (Fig. 4c), whereas Gata4^{hi} cells expressed transcription factors associated with the development of the columnar epithelium, including Gata6²¹, Pdx1²⁵, and Hnf4a²⁶

~

5. Offering some parallels between the embryonic mouse stomach epithelium to Barrett's esophagus has its origins in the manuscript by Wang et al. in 2011. It is an interesting but very speculative possibility. Without experimentation in adult mice and human organoids, this part of the manuscript is not on very solid ground.

Response

We agree the reviewer's criticism. We toned down the statement in the manuscript as follows:

~ **Introduction** A previous study suggested that the embryonic pseudostratified epithelium resides in adult TE, and also proposed the possibility that the residual embryonic cells are the origin of Barrett's metaplasia¹².

~Result Together, these findings support the notion that primitive TE cells and activation of MAPK/ERK signaling are associated with the pathogenesis of Barrett's metaplasia.

Minor:

1. Explanation/justification for the mouse stomach SCJ as a parallel for human gastro-esophageal junction needs to be provided.

Response

According to the reviewer's suggestion, we revised the manuscript as follows:

~ Different from mouse, the SCJ is not located in the stomach but at the esophageal-gastric junction in human^{12,13}. However, the human SCJs have common structure with the mouse SCJs in that KRT7⁺ TE cells exist between squamous and columnar epithelium^{13,28}.

2. In Fig 1b: At E13.5, KRT7+ cells seem to be present in different areas of stomach epithelium, and not as a single layer described by authors.

Response

KRT7 expression was restricted to SCJ at E18.5 but was broadly detected in the stomach epithelium (both proximal and distal stomach) at E13.5 (revised Fig.1b, 4c).

Whole stomach epithelium at E13.5 is "pseudo"stratified epithelium, which consists of a single layer epithelium. To avoid confusion, we revised the text as follows:

~ Both P63⁺ proximal stomach epithelium and GATA4⁺ distal stomach epithelium at E13.5 shape pseudostratified structures with expressing KRT7 (Fig. 1b), histologically similar to KRT7⁺ KRT14⁺ TE at E18.5 as previously reported^{12,13}.

3. In Extended Fig 1c, KRT7 expression is difficult to see and interpret. It seems to be on the surface of the stratified epithelium and also in stroma.

Response

Thanks to the reviewer's point, we revised the figure (revised Extended Data Fig. 1c). We demonstrated that KRT7 is expressed in both KRT14⁺ and KRT14⁻ proximal stomach epithelium but not in CLDN18⁺ distal stomach epithelium at E15.5.

4. Line 99-100: Without KRT7 lineage tracing Fig. 1a, b suggests, not shows, "differentiation" of KRT7+GATA+ cells into KRT7- GATA4+ CLDN18+ 100 columnar epithelium by E18.5.

Response

As the reviewer pointed, we revised the text as follows:

~ In the distal stomach, KRT7⁺ TE prominently expressed GATA4 at E11.5 and E13.5 (Fig. 1b). By contrast, the distal stomach epithelium was covered by the KRT7⁺ GATA4⁺ CLDN18⁺ columnar epithelium at E15.5 and E18.5 (Fig. 1a, Extended Data Fig. 1c), suggesting that KRT7⁺ GATA4⁺ TE cells differentiate into columnar epithelium.

5. Perhaps authors need to provide a more thorough review of recent (and quite plentiful) literature on the SCJ.

Response

We thank for this reviewer's helpful comments, and referred to more reviews and articles on SCJs.

Reviewers' Comments:

Reviewer #1:

Remarks to the Author:

The authors have addressed this reviewer's comments.

A few minor issues are listed as following,

1. Line 108. "Accordingly, KRT7+ TE may be the primitive epithelial type in the whole stomach". Can the author elaborate on this sentence? What does primitive mean here?
2. Line 128. Sox2 is limited to proximal stomach at E18.5. However, from the image it appears that some cells in the hindstomach remain positive for Sox2 at E18.5 (Fig1C). This is consistent with the expression of Sox2 in the crypts of hindstomach in adults (e.g. Arnold et al., cell stem cell, 2011). Authors may want to clarify.
3. Fig1. The labels should not use capital letters.
4. Line 130-131, the authors may want to specifically mentioned co-staining of Xgal and krt7, p63 etc in GATA4-CreER;R26lacZ when referring to Extended fig1e. This is important information.

Reviewer #3:

Remarks to the Author:

Authors have adequately addressed all main concerns.

Response to referee's comments

Reviewer #1:

We would like to thank the reviewer for his/her time and suggestions on helping us improve our manuscript. Our responses to each point are listed. The reviewers' comments are underlined.

The authors have addressed this reviewer's comments.

A few minor issues are listed as following.

1. Line 108. "Accordingly, KRT7+ TE may be the primitive epithelial type in the whole stomach". Can the author elaborate on this sentence? What does primitive mean here?

Response

According to the reviewer's comment, we revised as follows: "*Accordingly, KRT7⁺ transitional epithelium might be the primitive epithelial type harboring bidirectional differentiation potential into squamous and columnar epithelium in the developing stomach.*"

2. Line 128. Sox2 is limited to proximal stomach at E18.5. However, from the image it appears that some cells in the hindstomach remain positive for Sox2 at E18.5 (Fig1C). This is consistent with the expression of Sox2 in the crypts of hindstomach in adults (e.g. Arnold et al., cell stem cell, 2011). Authors may want to clarify.

Response

According to the reviewer's comment, we revised as follows:

"SOX2 expression gradually formed a proximal–distal gradient in the stomach epithelium from E11.5 to E13.5, and was largely downregulated in the distal stomach at E18.5 (Fig. 1c)."

3. Fig1. The labels should not use capital letters.

Response

We changed them to lowercase letters.

4. Line 130-131, the authors may want to specifically mentioned co-staining of Xgal and

krt7, p63 etc in GATA4-CreER;R26lacZ when referring to Extended fig1e. This is important information.

Response

According to the reviewer's comment, we revised as follows:

“By contrast, GATA4 expression shaped the distal–proximal gradient from E9.5 to E11.5, and was subsequently restricted to the distal stomach after E15.5 (Fig. 1c, d). Notably, GATA4-expressing cells retained the differentiation potential into P63⁺ basal cells at earlier stages (E9.5~E13.5) but not at E15.5 (Supplementary Fig. 1e).”